# Anisotropic mechanical properties of Dodecanophene nanosheets with pre-existing cracks by molecular dynamics simulation: Uncovering orientation- and temperature-induced variations

Wei Li *

Longnan Normal University, Longnan, China

* lnsfxylw@sina.com

## Abstract

This work presents a comprehensive molecular dynamics simulation study investigating the anisotropic mechanical response and fracture mechanisms of defective Dodecanophene nanosheets, a novel two-dimensional carbon allotrope. Using the AIREBO-M reactive force field validated against Density Functional Theory (DFT) calculations, we systematically evaluate the effects of crack orientation (0°–90°), temperature-dependent behavior (200–1000 K), and pre-existing crack size (30–60 Å) on elastic modulus, tensile strength, fracture toughness, and energy absorption. The nanosheets exhibit clear anisotropy: the y-direction shows higher stiffness (562.41 GPa) and strength (148.38 GPa), while the x-direction shows superior toughness (34.53 GPa). Crack orientation plays a critical role, with perpendicular cracks causing severe degradation (48.0–54.0%) compared to moderate losses (16–24%) for parallel cracks. Temperature-dependent behavior is pronounced, as toughness rises 160% at 200 K but declines 65.0% at 1000 K. Increasing pre-existing crack length drastically reduces strength (75.0–86.0%) and toughness (79.0–86.0%). Distinct failure modes emerge: x-loading promotes ductile behavior with crack deflection and gradual bond breaking, while y-loading induces brittle catastrophic fracture with rapid crack propagation. This represents the first systematic investigation of pre-existing crack effects on Dodecanophene's fracture mechanics across extreme thermal conditions (200–1000 K), providing critical insights for defect-tolerant design of 2D carbon materials.

## 1. Introduction

The discovery of graphene in 2004 by Geim and Novoselov [1] marked a revolutionary milestone in materials science, opening unprecedented avenues for two-dimensional (2D) carbon allotropes research. This single layer of carbon atoms

**Data availability statement:** All relevant data are within the manuscript.

**Funding:** The author(s) received no specific funding for this work.

**Competing interests:** The authors have declared that no competing interests exist.

arranged in a hexagonal lattice showed extraordinary mechanical properties, including exceptional tensile strength exceeding 130 *GPa*, Young's modulus approaching 1 *TPa*, and remarkable fracture strain of approximately 25% [2]. The exceptional properties of graphene stem from its perfect $sp^2$ hybridized carbon network, where each carbon atom forms three in-plane σ bonds with neighboring atoms, creating a robust planar structure [3]. The strong covalent bonding within the graphene plane contributes to its superior mechanical strength, while the delocalized π electrons provide excellent electrical conductivity [4].

Building upon the success of graphene, the scientific community has explored numerous other 2D carbon allotropes with distinct structural arrangements and properties. Graphyne, characterized by acetylene linkages connecting benzene rings, represents one of the most studied alternatives, exhibiting tunable band gaps and unique mechanical properties [5]. Similarly, graphdiyne, composed of diacetylene linkages, shows exceptional electrical properties and potential applications in energy storage systems [6]. Other notable 2D carbon structures include phagraphene, featuring four-, five-, six-, and seven-membered rings [7], penta-graphene with purely pentagonal rings [8], and biphenylene networks combining four- and six-membered rings [9]. The structural diversity among 2D carbon allotropes directly influences their mechanical behavior, with each configuration exhibiting unique responses to applied stresses. T-graphene, characterized by tetragonal symmetry, shows anisotropic mechanical properties with direction-dependent elastic moduli [10]. Haeckelite structures, incorporating five- and seven-membered rings, exhibit lower stiffness compared to graphene but offer enhanced flexibility [11]. These variations in mechanical properties arise from differences in bond angles, ring strain, and local atomic environments, highlighting the structure-property relationships governing 2D carbon materials.

Extensive research has been conducted on the mechanical properties and fracture behavior of various 2D carbon allotropes using both experimental and computational approaches. Molecular dynamics simulations have proven particularly valuable for investigating atomic-scale deformation mechanisms and crack propagation in these materials [12]. Studies on graphene have revealed that crack propagation follows preferential crystallographic directions, with crack deflection and branching occurring under specific loading condition [13]. The presence of defects, including vacancies, Stone-Wales defects, and grain boundaries, significantly influences the mechanical response and fracture resistance of graphene [14].

Research on graphyne and its derivatives has showed that the presence of acetylene linkages creates weaker bonds compared to the pure $sp^2$ network in graphene, resulting in lower mechanical strength but potentially enhanced flexibility [15,16]. A first-principles study investigated the effects of strain on the energetic and electronic properties of graphdiyne. Elastic parameters were derived from total energy calculations, revealing that graphdiyne is mechanically softer than graphyne due to its lower density of C–C bonds. The band gap of graphdiyne exhibits strain tunability, showing a monotonic increase with applied tensile strain. This behavior was attributed to reduced orbital overlap between carbon atoms as interatomic distances increase

under strain [17]. Molecular dynamics simulations reveal that α-graphyne and α2-graphyne exhibit anisotropic mechanical properties, with Young's modulus and fracture strain depending on the crystallographic orientation and structure type, where α2-graphyne shows higher stiffness but lower fracture strain than α-graphyne. While Young's modulus is size-independent, fracture strain and maximum stress decrease with increasing nanosheet dimensions, and both materials display unusually high Poisson's ratios, reaching up to 0.9 for certain sizes [18].

Temperature effects on the mechanical properties of 2D carbon materials have been extensively investigated, revealing significant dependencies on thermal conditions. Elevated temperatures generally lead to reduced elastic moduli and ultimate strengths due to increased atomic vibrations and thermal expansion [19]. Conversely, low temperatures enhance mechanical properties by reducing thermal motion and increasing bond stiffness [20]. These temperature dependencies are particularly important for applications in extreme environments and influence the design of devices operating under varying thermal conditions.

The influence of crack orientation on fracture behavior has been systematically studied across various 2D carbon allotropes. Kona et al. [21] employed molecular dynamics simulations to investigate the mechanical behavior and fracture mechanisms of penta-graphene, revealing that void defects at $sp^3$-hybridized carbon sites significantly reduce mechanical strength, while crack propagation involves bond reconstructions forming hexagonal and octagonal rings that enhance energy dissipation. The study showed that penta-graphene exhibits a higher critical stress intensity factor than graphene, especially under mode I loading, due to its ability to undergo out-of-plane deformation that delays crack initiation and promotes exceptional fracture toughness, with crack paths showing branching even under pure mode I conditions. Jin et al. [22] employed a non-equilibrium molecular dynamics framework with the AEBRO-M potential to study the fracture behavior of R12-graphene nanosheets containing a central crack under uniaxial loading, systematically analyzing mechanical properties—such as elastic modulus, strength, fracture strain, and stress intensity factor—across varying crack orientations (0°–90°), lengths (30–60 Å), and temperatures (200–1000 K). The results reveal pronounced anisotropy and temperature-dependent ductility, with Y-direction cracks showing greater brittleness and degradation, while a 90° crack promotes Mode I opening and stress redistribution, and lower temperatures (200 K) induce localized microplasticity that enhances fracture strain by ~5% compared to defect-free sheets despite reduced ultimate stress. Li and Zhang [23] investigated crack branching in graphene under complex stress conditions using molecular dynamics simulations and boundary-layer models, demonstrating that the maximum energy release rate (MERR) criterion, combined with a Wulff-like curve reflecting anisotropic fracture toughness, accurately predicts crack kinking directions—preferentially at 0°, 30°, and 60°—which align with the weaker zigzag fracture edges rather than the predictions of the maximum circumferential stress criterion. They further show that the sign change of the T-stress along the crack front, determined via the over-deterministic method, serves as an indicator for crack kinking, particularly in zigzag-oriented cracks, thereby enhancing the precision of nanocrack path prediction in graphene.

The current study focuses on investigating the mechanical properties and fracture behavior of Dodecanophene, a recently proposed 2D carbon allotrope characterized by a unique periodic arrangement of four-, six-, and eight-membered rings [24,25]. This distinctive structural framework creates an anisotropic material with potentially tunable mechanical properties that differ significantly from graphene and other well-studied carbon allotropes. The research aims to comprehensively characterize the elastic modulus, ultimate tensile strength, fracture toughness, and energy absorption capacity of Dodecanophene under various loading conditions. Specifically, the investigation examines the effects of crack orientation, temperature variation, and defect size on the mechanical response, providing fundamental insights into the structure-property relationships governing this novel carbon material. Through systematic molecular dynamics simulations, this work seeks to establish the mechanical property database for Dodecanophene and evaluate its potential for engineering applications requiring high-performance 2D materials with tailored mechanical characteristics.

The recent work by Zhang and Wei [25] provides a valuable foundational study on the mechanical properties of Dodecanophene nanosheets, focusing on the influence of size, temperature, and defect concentration on key metrics like

Young's modulus, ultimate stress, and toughness. Their use of non-equilibrium molecular dynamics (NEMD) to characterize the material's anisotropic elastic and strength properties across varying sizes and temperatures (up to 700 K) establishes a crucial baseline. However, a critical gap remains in the understanding of Dodecanophene's fracture mechanics, particularly concerning the behavior of pre-existing structural flaws. Our current study is specifically designed to address this by moving beyond the general effects of point defects and size. We employ molecular dynamics simulations to systematically investigate the complex interplay between pre-existing crack geometry (orientation), defect size (crack length), and temperature (up to 1000 K) on the material's fracture toughness and energy absorption capacity. This unique focus on crack-tip mechanics and the resulting anisotropic fracture behavior under varying crack orientations and extreme thermal conditions (including higher temperatures not covered in previous work) provides a distinct and essential contribution, offering the necessary insights for reliable engineering applications of Dodecanophene where structural integrity and flaw tolerance are paramount.

While extensive research has characterized the mechanical properties of various two-dimensional carbon allotropes, including graphene, graphyne, and other novel structures, a comprehensive understanding of the fracture mechanics of the newly proposed Dodecanophene nanosheet remains elusive. The existing literature on 2D materials often focuses on generalized defect effects or the elastic regime under ambient conditions. Our study transcends these limitations by providing the first systematic investigation into the anisotropic fracture behavior of Dodecanophene, specifically under the combined influence of pre-existing crack geometry (orientation), defect size (crack length), and a wide range of temperatures (200–1000 K). By focusing on the critical stress intensity factor, fracture toughness, and energy absorption capacity, this work establishes a crucial mechanical property database for Dodecanophene, offering unprecedented atomic-level insights into the material's flaw tolerance and failure modes. This unique and comprehensive approach is essential for evaluating Dodecanophene's potential for reliable engineering applications where structural integrity under extreme thermal and mechanical loading is paramount.

## 2. Computational methodology

Classical molecular dynamics simulations were conducted to examine the mechanical response of Dodecanophene using the LAMMPS computational framework [26]. The selection of appropriate interatomic potential functions represents a fundamental consideration in atomistic modeling, as these functions dictate the accuracy of predicted material behavior. For carbon-based nanomaterials, several empirical potentials have showed effectiveness, including the Reactive Empirical Bond Order (REBO) formulation [27], the Adaptive Intermolecular Reactive Empirical Bond Order (AIREBO) extension [28], and reactive force fields such as ReaxFF [29]. These approaches offer varying degrees of computational efficiency and physical fidelity.

The present investigation utilized the AIREBO-M potential [30], which incorporates modifications specifically designed to enhance the description of graphitic carbon systems. Temporal integration employed the velocity-Verlet scheme [31], selected for its superior energy conservation properties and numerical stability throughout extended simulation periods. System equilibration followed a dual-stage protocol to ensure thermal and mechanical equilibrium. An initial equilibration period of 1 nanosecond was performed under NVT conditions (fixed particle number, volume, and temperature), with temperature regulation achieved through Nose–Hoover thermostatting. Subsequently, a 3-nanosecond equilibration under NPT conditions (fixed particle number, pressure, and temperature) was implemented, employing Nose–Hoover coupling for both thermal and barostat control. All calculations utilized a 1 femtosecond integration timestep, with periodic boundary conditions applied across all spatial directions to simulate infinite bulk behavior. Mechanical loading is applied in the form of uniaxial tensile strain at a constant engineering strain rate of $10^5\ s^{-1}$ along the longitudinal axis of the nanosheet [32].

Tensile testing simulations involved constraining one boundary of the Dodecanophene nanosheet while applying controlled deformation to the opposing edge at constant strain rate. This configuration enabled comprehensive examination of deformation physics, stress evolution, and failure mechanisms under uniaxial loading conditions. Atomic-scale

visualization and post-processing utilized OVITO software (version 3.7.3) [33], facilitating real-time monitoring of bond dissociation events and stress concentration phenomena. Stress calculations employed the virial theorem formulation of the Cauchy stress tensor, incorporating both kinetic and configurational contributions. The mathematical expression for stress components follows [34]:

$$\sigma_{ij} = \frac{1}{V}\left(\sum_k m^{(k)} v_i^{(k)} v_j^{(k)} + \sum_{k<l} r_i^{(kl)} f_j^{(kl)}\right)$$

(1)

where $V$ is the system volume, $m^{(k)}$ is the mass of atom $k$, $v_i^{(k)}$ denotes the $ith$ component of the velocity vector of atom $k$, $r_i^{(kl)}$ represents the separation between atoms $k$ and $l$ along the $ith$ direction, and $f_j^{(kl)}$ is the $jth$ component of the interatomic force between atoms $k$ and $l$.

This stress formulation enables accurate determination of both local and global stress states during mechanical loading, particularly near crack tips and defect sites. Elastic modulus determination involved analysis of the initial linear portion of stress-strain curves, calculating the derivative [35]:

$$E = \frac{\partial \sigma}{\partial \in}$$

(2)

Linear regression analysis was applied to stress-strain data within the initial 3% strain range to extract the elastic modulus, ensuring evaluation occurs within the linear elastic regime prior to nonlinear deformation onset. Fracture analysis focused on Mode I crack propagation, characterized by tensile loading perpendicular to the crack plane. The critical stress intensity factor, representing fracture toughness was determined using [36]:

$$K_{IC} = \sigma_f \sqrt{2htan\left(\frac{\pi a}{2h}\right)}$$

(3)

where $\sigma_f$ represents the ultimate tensile strength, $2h$ denotes the nanosheet width, and $2a$ indicates the initial crack length. This formulation accounts for geometric effects of crack dimensions and specimen geometry on stress singularity at crack tips, providing quantitative assessment of fracture resistance under Mode I loading. The integrated computational approach, combining atomistic visualization, stress analysis, and fracture mechanics principles, provides a comprehensive framework for evaluating mechanical behavior and failure characteristics of defective Dodecanophene nanosheets. This methodology enables detailed investigation of how structural defects and loading conditions influence the mechanical response of two-dimensional carbon materials.

The structural architecture of Dodecanophene is depicted in Fig 1, which reveals the distinctive two-dimensional carbon network composed of three types of polygonal rings arranged in a periodic pattern. The structure exhibits a regular tessellation of four-membered rings (cyclobutane-like squares), six-membered rings (benzene-like hexagons), and twelve-membered rings that form a continuous planar lattice. The visualization employs a ball-and-stick model where carbon atoms are represented as gray spheres connected by cylindrical bonds, providing clear visualization of the atomic connectivity and ring topology. The planar nature of the structure is evident from the uniform z-coordinate positioning of all atoms, confirming the two-dimensional character of this carbon allotrope.

Fig 2 presents the atomic-scale configurations of Dodecanophene nanosheets employed in the crack angle investigation, systematically illustrating how crack orientation affects the mechanical response under uniaxial tensile loading. Each panel depicts a 150 × 150 Å² nanosheet containing a pre-existing edge crack with consistent dimensions (40 Å length, 8 Å width) but varying angular orientations with respect to the applied loading direction. The crack angle parametric study encompasses five distinct orientations: 0°, 30°, 45°, 60°, and 90°, where the angle is measured between the crack plane

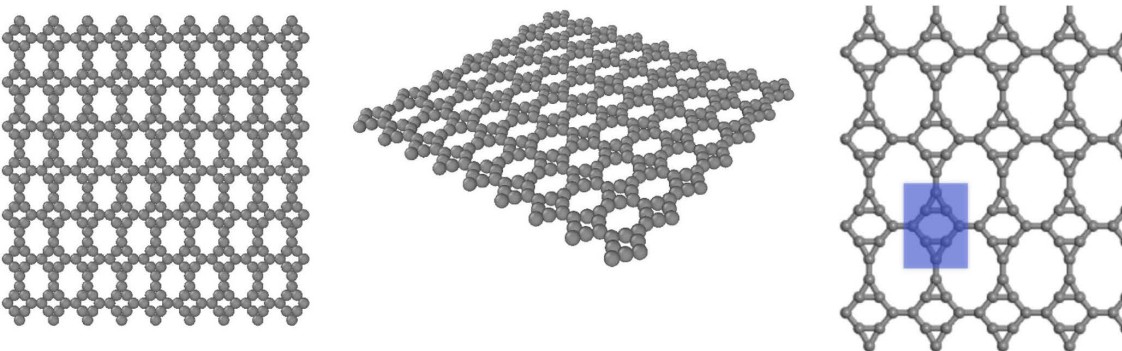

**Fig 1. Atomic structure of two-dimensional Dodecanophene, displaying the characteristic periodic arrangement carbon rings.** The ball-and-stick representation illustrates the planar geometry and connectivity pattern of carbon atoms (gray spheres) linked by covalent bonds, highlighting the unique topological framework that distinguishes Dodecanophene from other carbon allotropes.

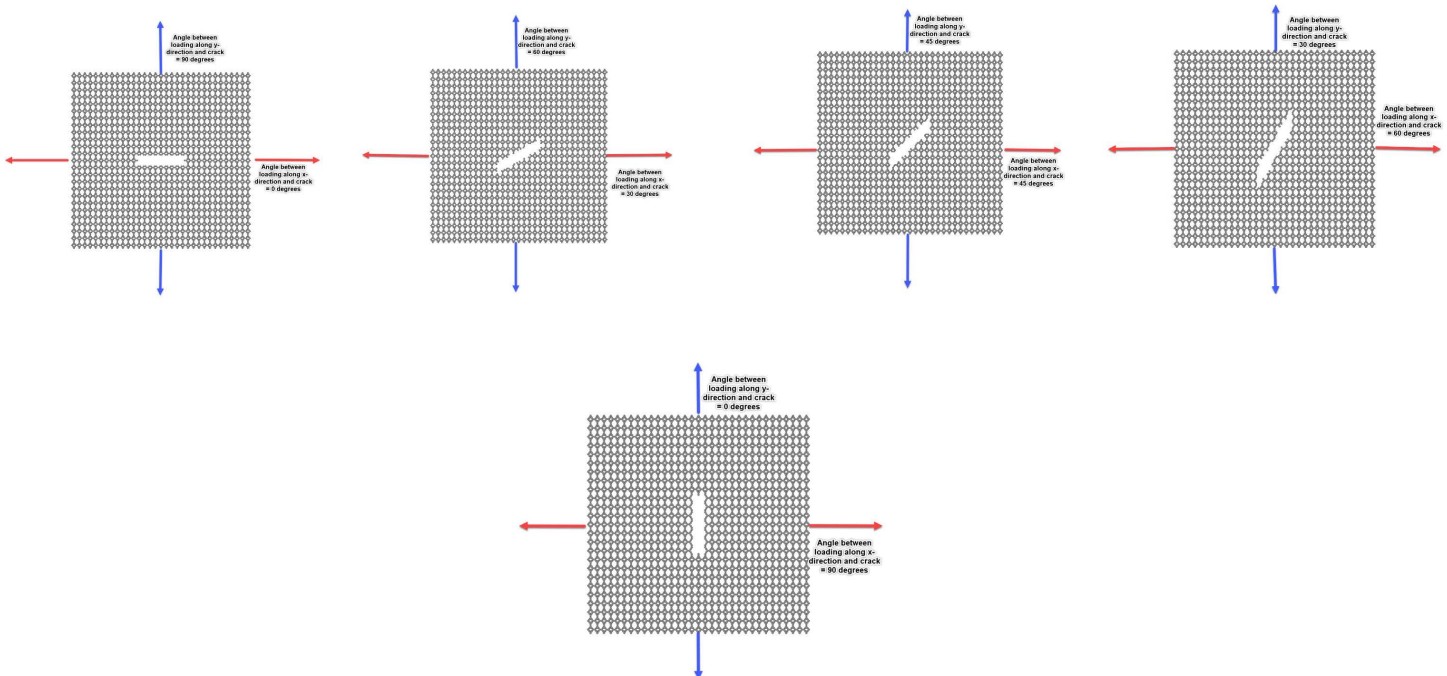

**Fig 2. Atomic configurations of Dodecanophene nanosheets (150 × 150 Å²) containing pre-existing edge cracks oriented at different angles relative to the tensile loading direction.**

and the direction perpendicular to the applied tensile stress. This systematic variation enables comprehensive characterization of the angular dependence of fracture properties, including elastic modulus degradation, ultimate strength reduction, and fracture toughness variation.

The 0° configuration represents the case where the crack is aligned parallel to the loading direction, minimizing direct tensile stress across the crack plane. This orientation typically exhibits the least severe stress concentration effects and shows the material's resistance to crack opening under parallel loading conditions. In the 90° configuration, the crack is oriented perpendicular to the loading direction, representing a classical Mode I fracture scenario where tensile stresses

act normal to the crack plane. This orientation typically produces maximum stress concentration at the crack tip and represents the most critical loading condition for crack propagation initiation. The intermediate angles (30°, 45°, and 60°) introduce mixed-mode loading conditions where both normal and shear stresses contribute to crack tip stress fields. These oblique orientations create complex stress distributions that can influence both crack initiation thresholds and propagation trajectories, often resulting in crack deflection or branching behaviors.

Each atomic configuration maintains identical boundary conditions and nanosheet dimensions to ensure that observed variations in mechanical properties are solely attributable to crack orientation effects rather than geometric or size-related factors. The consistent crack dimensions across all orientations enable direct comparison of angular effects on fracture behavior. The visualization employs atomic-scale resolution to clearly delineate the crack geometry and its relationship to the underlying Dodecanophene lattice structure. The crack edges are precisely defined at the atomic level, ensuring accurate representation of the stress concentration regions that govern fracture initiation and propagation processes. This systematic approach to crack orientation studies provides fundamental insights into the directional dependencies of fracture behavior in Dodecanophene, revealing how the anisotropic structural characteristics of this carbon allotrope influence its mechanical response under various loading configurations.

All crack simulations were indeed performed at 300 K, unless otherwise specified in the sections where temperature effects were explicitly investigated.

The application of linear elastic fracture mechanics (LEFM) and the stress intensity factor ($K_{IC}$) to nanoscale atomic systems requires careful consideration of underlying assumptions and inherent limitations. Classical LEFM was originally developed for macroscale continuum materials under the assumptions of material continuity, isotropy, homogeneity, and purely elastic behavior up to the point of fracture initiation [37]. In atomic lattices such as Dodecanophene, these assumptions are not strictly satisfied due to the discrete nature of atoms, directional bonding, lattice trapping effects, and atomic-scale bond breaking and reformation processes [38].

At the nanoscale, crack propagation is governed by sequential bond rupture rather than continuous stress field singularities, and the crack tip experiences lattice resistance (Peierls barrier) that impedes smooth advancement [39]. Furthermore, the classical $K_{IC}$ formulation (Equation 3) is rigorously applicable only to Mode I (tensile opening) fracture where the crack plane is perpendicular to the applied load. For oblique crack orientations (30°, 45°, 60°), mixed-mode loading conditions arise, involving both Mode I (opening) and Mode II (shear) components, which cannot be fully captured by a single stress intensity factor without decomposition into $K_I$ and $K_{II}$ components.

Despite these theoretical limitations, the stress intensity factor has been widely employed in molecular dynamics studies of two-dimensional materials as an *effective* fracture parameter that enables meaningful comparative analysis across different crack geometries, loading conditions, and material systems [36,40–43]. Zhang et al. [44] demonstrated that while absolute $K_{IC}$ values from atomistic simulations may deviate from continuum predictions due to lattice discreteness, the *relative trends* and *scaling relationships* remain physically meaningful and consistent with experimental observations in graphene.

In the present study, we employ $K_{IC}$ as an effective fracture parameter with the explicit understanding that:

(1) The calculated values represent *effective* fracture toughness incorporating atomic-scale effects rather than pure continuum predictions;

(2) For mixed-mode loading cases (oblique crack angles), the $K_{IC}$ values represent *composite* measures of fracture resistance that combine Mode I and Mode II contributions, providing qualitative insights into crack-orientation effects rather than rigorous mode decomposition;

(3) The primary utility of $K_{IC}$ lies in *comparative analysis*—evaluating how crack angle, temperature, and defect size alter fracture resistance relative to pristine conditions and enabling benchmarking against other 2D carbon allotropes studied using similar methodologies;

(4) The observed trends in $K_{IC}$ with varying crack angles (Fig 7) reflect the physical reality that crack orientation relative to loading direction fundamentally alters the stress state at the crack tip and the propensity for crack propagation, even if the continuum singularity formulation is an approximation at atomic scales.

Therefore, while we acknowledge the theoretical limitations of applying classical LEFM to atomic systems, we maintain that $K_{IC}$ provides valuable quantitative insights into fracture behavior when interpreted as an effective parameter capturing the combined influences of crack geometry, loading mode, and atomic-scale energy dissipation mechanisms. All $K_{IC}$ values reported herein should be understood within this framework—as comparative measures enabling systematic evaluation of fracture trends rather than absolute predictions of continuum fracture toughness.

## 3. Results

### 3.1. Mechanical properties of pristine Dodecanophene

The stress-strain curves reveal pronounced mechanical anisotropy in pristine Dodecanophene (see Fig 3), with significantly different deformation behaviors along the two principal directions. In the y-direction, the material exhibits a higher elastic modulus of 562.41 $GPa$ compared to 387.18 $GPa$ in the x-direction, indicating greater stiffness perpendicular to the molecular chain direction. The y-direction also shows superior ultimate stress capacity (148.38 $GPa$ vs. 92.78 $GPa$), suggesting stronger intermolecular interactions and resistance to failure. However, the x-direction shows considerably higher

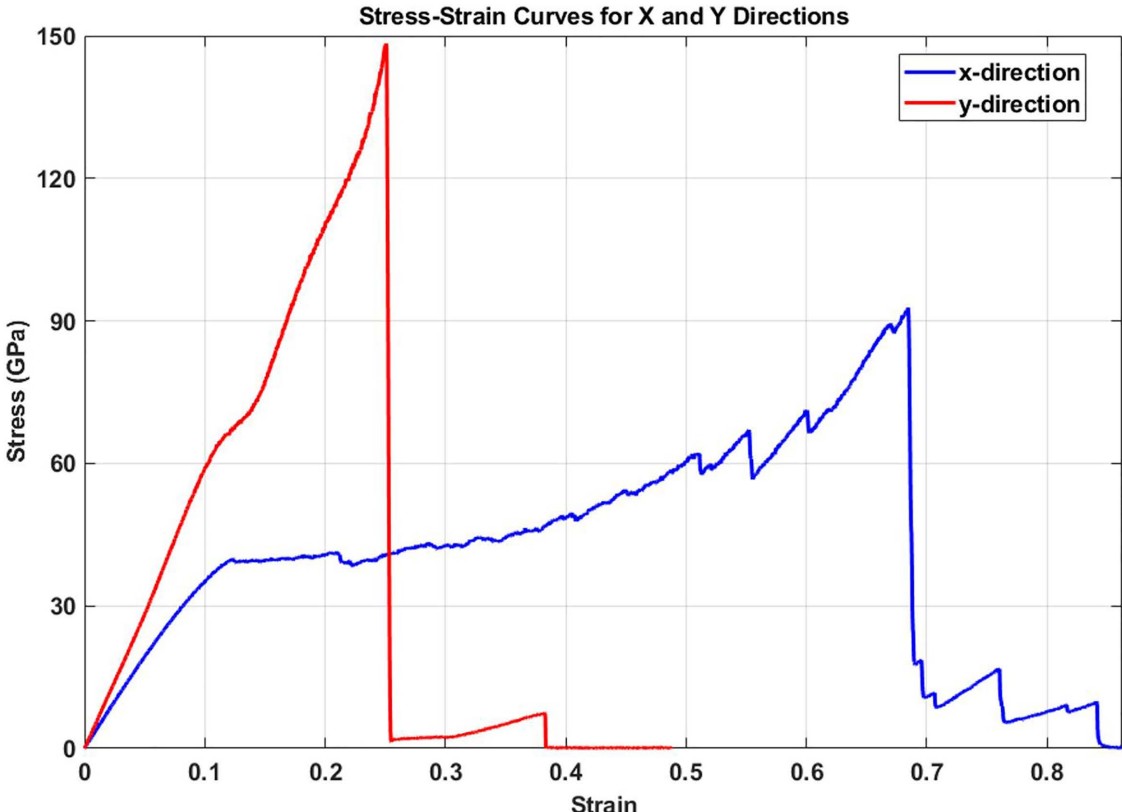

**Fig 3. Stress-strain curves for pristine Dodecanophene under uniaxial tensile loading in the x-direction (blue) and y-direction (red), showing distinct mechanical anisotropy and failure mechanisms.**

toughness (34.53 vs. 18.15), indicating greater energy absorption capacity before failure, which is attributed to the more gradual stress increase and extended plastic deformation region. The sharp stress drop in the y-direction at approximately 0.25 strain suggests brittle failure, while the x-direction exhibits a more ductile response with progressive stress accumulation up to 0.67 strain before ultimate failure. This anisotropic behavior reflects the underlying molecular structure and bonding characteristics of dodecanophene, where mechanical properties are strongly dependent on the loading direction relative to the molecular orientation.

## 3.2. Comparison and validation of elastic modulus with DFT results

To validate the elastic modulus results obtained in this investigation, a comparative analysis was conducted against previously reported computational findings for Dodecanophene. The current molecular dynamics (MD) study, detailed in Section 3.1 and illustrated in Fig 3, determined the elastic modulus for a pristine Dodecanophene nanosheet at 300 $K$ to be 387.18 $GPa$ along the x-direction and 562.41 $GPa$ along the y-direction. These values show favorable agreement with prior density functional theory (DFT) calculations, as shown in Table 1.

The comparison reveals that the MD simulation results are in close proximity to the established DFT values, with a maximum deviation of 4.5% in the x-direction and a minimum deviation of 1.0% in the y-direction. The slight discrepancies are attributable to the inherent differences between the all-electron DFT method and the empirical potential (AIREBO-M) used in the MD simulations. Specifically, the AIREBO-M potential, while efficient for large-scale simulations, provides an approximation of the interatomic forces, which can lead to minor variations in the calculated elastic constants compared to the more rigorous DFT approach. The high degree of correlation, particularly in the y-direction, confirms the reliability of the computational methodology and parameters employed in this study for accurately modeling the mechanical response of Dodecanophene nanosheets.

## 3.3. Effect of crack angle

Fig 4 depicts the stress-strain response of pristine Dodecanophene nanosheets under uniaxial tensile loading in both (a) x-direction and (b) y-direction at 300 K, with a nanosheet size of 150 × 150 Å². The curves show the fundamental anisotropic mechanical properties of the material, revealing distinct differences in strength, stiffness, and ductility between the two primary crystallographic directions. This anisotropy arises from the unique atomic configuration of Dodecanophene. The stress-strain curves exhibit classical elastic-plastic behavior with distinct regions corresponding to elastic deformation, yielding, strain hardening, and ultimate failure. In both directions, the initial linear portion represents the elastic regime where stress is proportional to strain, with the slope defining the elastic modulus. The deviation from linearity marks the onset of plastic deformation, where atomic bonds begin to stretch beyond their equilibrium positions and defects start to nucleate within the crystal structure.

The x-direction curve (Fig 4a) shows an elastic modulus of approximately 387.18 $GPa$ and reaches an ultimate tensile stress of 92.78 $GPa$. The material exhibits significant strain hardening after yielding, indicating its ability to accommodate large deformations through bond stretching and local structural rearrangements. The total strain to failure extends beyond 0.3, demonstrating considerable ductility in this direction. The gradual increase in stress after yielding suggests that the material can redistribute loads effectively through its atomic network, allowing for substantial plastic deformation before catastrophic failure. In contrast, the y-direction curve (Fig 4b) shows superior mechanical properties with a higher elastic

**Table 1. Comparison of elastic modulus from current MD simulation with reference DFT results.**

| Parameter | Loading Direction | Current MD Result (GPa) | Reference DFT Value (GPa) | Percentage Error (%) |
|---|---|---|---|---|
| Elastic Modulus | x-direction | 387.18 | 405.20 [24] | 4.5% |
| Elastic Modulus | y-direction | 562.41 | 568.00 [24] | 1.0% |

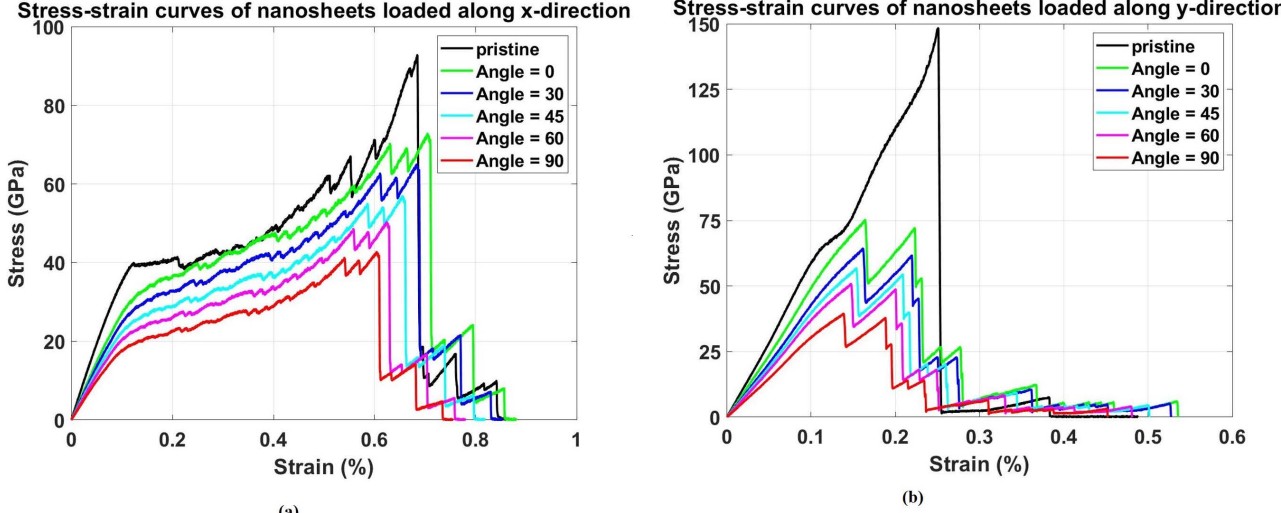

**Fig 4. Stress–strain response of Dodecanophene nanosheet under uniaxial tensile loading in (a) x-direction and (b) y-direction at 300 K, with a nanosheet size of 150×150 Å², highlighting the material's anisotropic mechanical properties.**

modulus of 562.41 *GPa* and an ultimate tensile stress of 148.38 *GPa*. However, the strain to failure is significantly lower compared to the x-direction, indicating reduced ductility despite higher strength. This behavior can be attributed to the orientation of the carbon ring structure relative to the loading direction, where stronger covalent bonds are more effectively aligned to resist deformation in the y-direction. The steeper slope and higher peak stress suggest that the atomic arrangement provides more efficient load transfer pathways along this crystallographic orientation.

The area under each stress-strain curve represents the material's toughness, with the x-direction showing a toughness of 34.53 *GPa* compared to 18.15 *GPa* in the y-direction. This indicates that while the y-direction provides superior strength and stiffness, the x-direction offers better energy absorption capacity due to its enhanced ductility. This trade-off between strength and toughness is characteristic of anisotropic materials and has important implications for applications where either high strength or high energy absorption is prioritized. The anisotropic behavior revealed in Fig 3 establishes the baseline mechanical properties that serve as reference points for evaluating the degradation effects of various defects, temperature variations, and other structural modifications studied throughout this research.

Fig 5 depicts the variation of elastic modulus with crack angle for Dodecanophene nanosheets under both x- and y-directional loading. The graph shows a monotonic decrease in elastic modulus as the crack angle increases from 0° to 90° for both loading directions. This behavior can be attributed to the anisotropic crystal structure of Dodecanophene that creates directional dependencies in mechanical response. When cracks are aligned parallel to the loading direction, the material maintains maximum stiffness as the stress field encounters the strongest atomic bonds. Comparing with pristine specimens, the elastic modulus shows significant degradation due to crack presence. For the x-direction, the pristine elastic modulus of 387.18 *GPa* decreases to 295.01 *GPa* (23.8% reduction) for 0° crack angle and 200.11 *GPa* (48.3% reduction) for 90° crack angle. In the y-direction, the pristine value of 562.41 *GPa* reduces to 471.91 *GPa* (16.1% reduction) for 0° crack angle and 294.71*GPa* (47.6% reduction) for 90° crack angle. These substantial reductions show that even relatively small cracks can dramatically compromise the material's stiffness.

Fig 6 illustrates the ultimate tensile stress as a function of crack angle for both loading directions. The graph reveals a systematic decline in ultimate stress with increasing crack angle, reflecting the progressive weakening of the material's

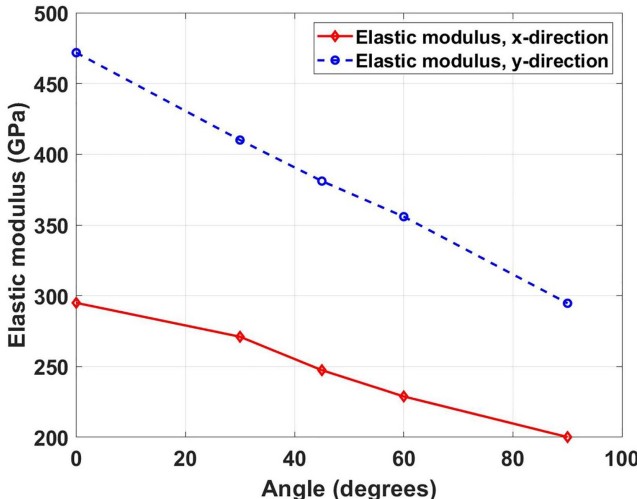

**Fig 5. Elastic modulus variation of Dodecanophene with crack angles (0° to 90°) under x- and y-directional tensile loading, for a 150×150 Å² nanosheet with a 40 Å crack at 300 K, showing the impact of crack orientation on stiffness.**

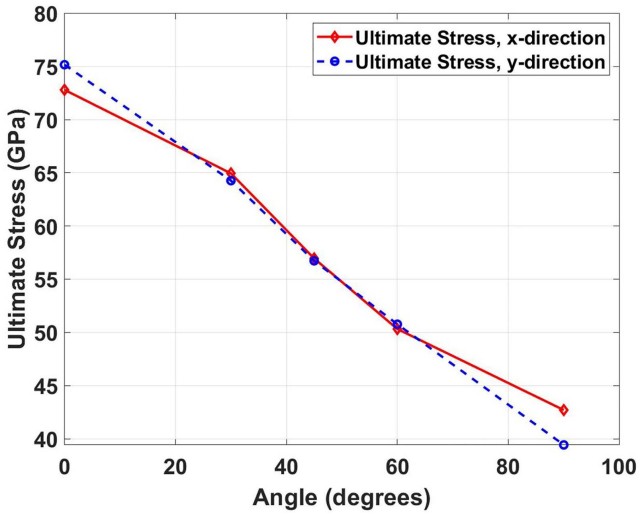

**Fig 6. Ultimate tensile stress of Dodecanophene as a function of crack angle (0° to 90°) under loading in x- and y-directions, for a 150×150 Å² nanosheet with a 40 Å crack at 300 K, illustrating the effect of crack orientation on strength.**

load-bearing capacity. This phenomenon occurs because crack orientation relative to the applied stress field determines the mode of stress concentration at the crack tip. At 90° crack angle, the crack opening is directly opposed by the tensile load. The comparison with pristine specimens shows dramatic strength reductions. In the x-direction, the pristine ultimate stress of 92.78 $GPa$ drops to 72.80 $GPa$ (21.5% reduction) for 0° cracks and 42.70 $GPa$ (54.0% reduction) for 90° cracks. The y-direction shows pristine ultimate stress of 148.38 $GPa$, which decreases to 75.16 $GPa$ (49.4% reduction) for 0° cracks and 39.42 $GPa$ (73.4% reduction) for 90° cracks. The more severe degradation in ultimate stress compared to elastic modulus indicates that crack presence has a more pronounced effect on the material's failure strength than on its initial stiffness.

Fig 7 represents the stress intensity factor ($K_{IC}$) variation with crack angle for both loading directions. The graph shows a systematic decrease in fracture toughness as the crack angle with the loading direction increases. This reduction indicates diminishing fracture resistance, with the physical explanation involving the changing stress state at the crack tip. When the crack is parallel to loading (0°), maximum tensile stress concentrates at the crack tip, requiring higher stress intensity for crack extension. As the angle increases, shear components become dominant, reducing the critical stress intensity needed for crack propagation. Comparing the $K_{IC}$ values (obtained using Eq. (3)), the 0° crack specimens show $K_{IC}$ values of 5.95 $MPa \cdot m^{0.5}$ (x-direction) and 6.14 $MPa \cdot m^{0.5}$ (y-direction), while 90° crack specimens exhibit significantly lower values of 3.49 $MPa \cdot m^{0.5}$ (x-direction) and 3.22 $MPa \cdot m^{0.5}$ (y-direction), representing reductions of 41.3% and 47.6%, respectively. The consistently higher $K_{IC}$ values in the y-direction suggest superior crack resistance along this crystallographic orientation.

Fig 8 presents the toughness variation with crack angle, showing the energy absorption capacity before failure. The graph exhibits the steepest decline among all mechanical properties as crack angle increases, reflecting the material's diminishing ability to absorb strain energy as the crack orientation becomes less favorable. The physical mechanism involves the progressive transition from mode I (opening) to mixed-mode fracture, where the material's intrinsic energy dissipation mechanisms become less effective. The toughness comparison reveals the most severe impact of crack presence on mechanical performance. For the x-direction, pristine toughness of 34.53 $GPa$ decreases to 33.24 $GPa$ (3.7% reduction) for 0° cracks and 16.71 $GPa$ (51.6% reduction) for 90° cracks. The y-direction shows pristine toughness of 18.15 $GPa$, which reduces to 13.10 $GPa$ (27.8% reduction) for 0° cracks and 5.80 $GPa$ (68.0% reduction) for 90° cracks. The relatively modest reduction for 0° cracks compared to the dramatic decrease for 90° cracks emphasizes the critical importance of crack orientation in determining energy absorption capacity.

The observed anisotropy in crack resistance, where the x-direction exhibits superior fracture toughness despite the y-direction possessing higher stiffness and strength, is directly attributable to the distinct atomic-scale failure mechanisms. The fracture toughness (a measure of crack resistance) is significantly higher in the x-direction compared to the y-direction. This phenomenon is rooted in the way the Dodecanophene lattice responds to stress in each orientation. Under x-direction loading, the stress concentration at the crack tip is relieved through a more ductile failure mode. This is

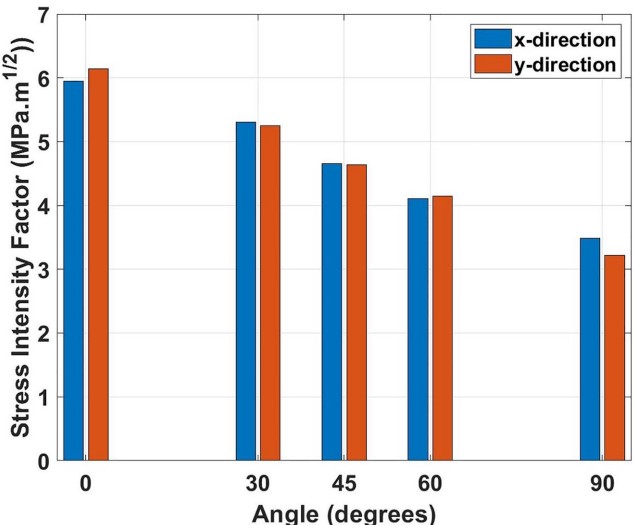

**Fig 7. Stress intensity factor ($K_{IC}$) of Dodecanophene versus crack angle (0° to 90°) under x- and y-directional loading, for a 150 × 150 Å² nanosheet with a 40 Å crack at 300 K, demonstrating the influence of crack orientation on fracture resistance.**

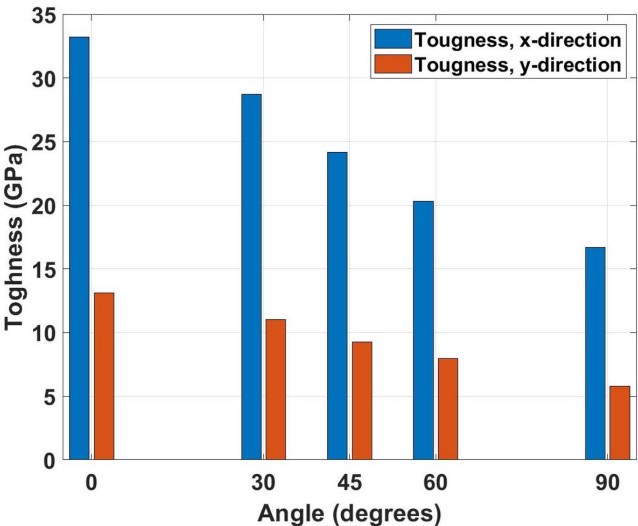

**Fig 8. Toughness of Dodecanophene as a function of crack angle (0° to 90°) under tensile loading in x- and y-directions, for a 150×150 Å²  nanosheet with a 40 Å crack at 300 K, showing the dependence of energy absorption on crack orientation.**

characterized by significant crack tip blunting, bond rearrangement, and crack deflection along the weaker structural paths of the nanosheet's unique four-, six-, and eight-membered ring arrangement. This gradual, energy-dissipative process requires substantially more energy for crack propagation, thus manifesting as superior fracture toughness.

Conversely, under y-direction loading, the material exhibits a brittle, catastrophic failure mode. The atomic structure along this axis facilitates a more direct and rapid cleavage path, leading to minimal crack tip blunting and an absence of significant bond rearrangement or crack deflection. The crack propagates rapidly in a straight line, resulting in lower energy absorption and, consequently, lower fracture toughness, despite the material's higher elastic modulus and ultimate strength in this direction. Therefore, the superior crack resistance of the x-direction is a direct consequence of its ability to promote energy-dissipative, ductile failure mechanisms at the crack tip.

### 3.4. Effect of temperature

Fig 9 presents the stress-strain curves at different temperatures. The temperature is considered in the range of 200 $K$ to 1000 $K$, with a 90° crack angle and 150×150 $\text{Å}^2$ nanosheet size. Fig 9 illustrates a systematic reduction in both the stress levels and the strain to failure as temperature increases, resulting in progressively smaller areas under the stress-strain curves. This indicates a dramatic decrease in energy absorption capacity with increasing temperature. The physical mechanism involves thermal activation of bond breaking processes and reduced strain hardening capability at elevated temperatures. Higher temperatures promote earlier onset of plastic deformation and reduce the material's ability to sustain large strains before failure. The toughness comparison shows the most severe temperature dependence among all properties. These extreme variations show that toughness is the most temperature-sensitive property, with low temperatures providing substantially enhanced energy absorption while high temperatures severely compromise the material's damage tolerance.

A notable observation from the stress-strain curves in Fig 9 is the slightly enhanced fracture strain of the pre-cracked Dodecanophene nanosheet (with a 90° crack) compared to the pristine structure when subjected to uniaxial tension at the low temperature of 200 $K$. This phenomenon, where a defect appears to enhance a ductility-related property, is counter-intuitive but possesses a rational physical explanation rooted in the mechanics of fracture at reduced thermal energy.

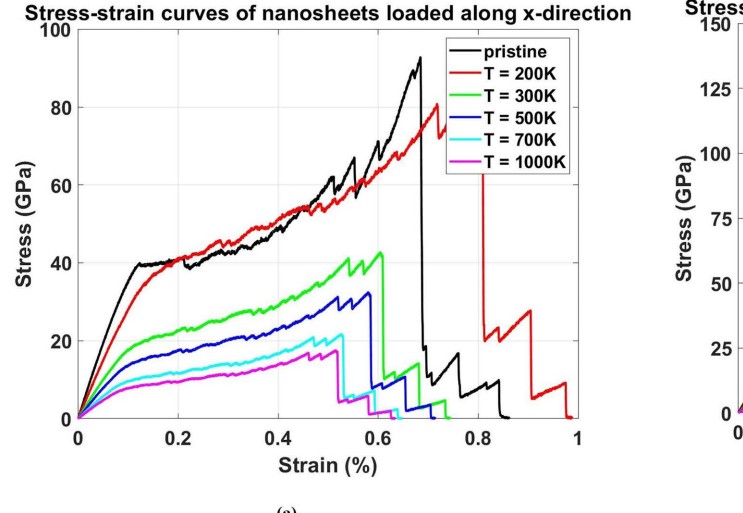
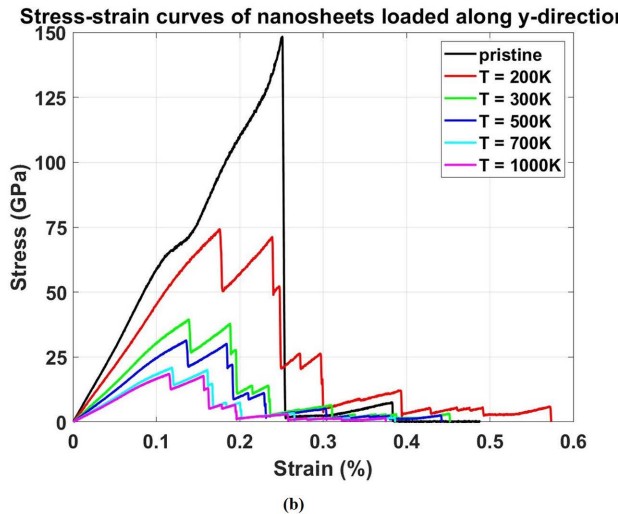

**Fig 9. Stress–strain curves of Dodecanophene nanosheet under uniaxial tensile loading in (a) x-direction and (b) y-direction across temperatures (200K to 1000 K), with a 90° crack angle and 150×150 Å² nanosheet size, revealing temperature-dependent mechanical behavior.**

At 200 $K$, the significantly reduced thermal energy restricts the overall atomic mobility and suppresses large-scale plastic deformation. However, the presence of the crack tip introduces a severe, highly localized stress concentration. This intense stress field, combined with the low temperature, can trigger a localized, non-uniform deformation mechanism known as microplasticity or crack-tip blunting. The restricted atomic motion at 200 $K$ forces the material's response to be highly concentrated at the crack tip, where the energy barrier for bond rearrangement is locally lowered by the intense stress. This localized energy dissipation mechanism, involving the rearrangement of carbon bonds (e.g., the formation of new rings or local structural relaxation) in the immediate vicinity of the crack tip, effectively blunts the crack. This blunting action temporarily stabilizes the crack, requiring a greater overall macroscopic strain to achieve the critical condition for catastrophic failure. This effect is often more pronounced at low temperatures, where the competition between brittle fracture (favored by low temperature) and localized ductile behavior (favored by crack-tip stress concentration) results in a temporary, localized increase in energy absorption capacity, thereby leading to a marginally higher overall fracture strain than the perfectly brittle failure of the pristine structure. This observation aligns with similar findings in other 2D materials, where restricted atomic motion at low temperatures fosters localized microplasticity near the crack tip, enhancing ductility.

Fig 10 depicts the variation of elastic modulus with temperature ranging from 200 $K$ to 1000 $K$ for Dodecanophene nanosheets under both x- and y-directional loading. The crack angle is considered as 90° and the nanosheet size is equal to 150×150 Å². The graph shows a nonlinear decrease in elastic modulus with increasing temperature, showing a steep initial decline followed by a more gradual reduction at higher temperatures. This behavior can be attributed to the thermal expansion of the lattice and increased atomic vibrations that weaken interatomic bonds. At elevated temperatures, the increased kinetic energy of atoms leads to bond softening and reduced resistance to deformation. The thermal degradation follows expected trends where higher temperatures progressively compromise the material's structural integrity. Comparing with pristine specimens at 300 $K$, the temperature effects are substantial. For the x-direction, the elastic modulus increases from 200.11 $GPa$ at 300 $K$ to 299.52 $GPa$ at 200 $K$ (49.7% increase) and decreases dramatically to 94.86 $GPa$ at 1000 $K$ (52.6% reduction). In the y-direction, the modulus rises from 294.71 $GPa$ at 300 $K$ to 435.10 $GPa$ at 200 $K$ (47.6% increase) and drops to 167.97 $GPa$ at 1000 $K$ (43.0% reduction). These variations show the critical importance of temperature control in applications requiring high stiffness.

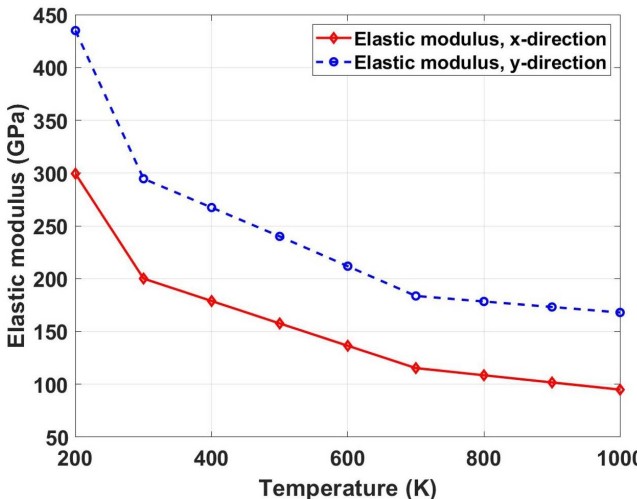

**Fig 10. Elastic modulus of Dodecanophene versus temperature (200 K to 1000 K) under x- and y-directional loading, with a 90° crack angle and 150×150 Å² nanosheet size, illustrating thermal effects on material stiffness.**

Fig 11 represents the temperature-dependent evolution of ultimate tensile stress in dodecanophene nanosheets obtained through molecular dynamics (MD) simulations, demonstrating the thermomechanical response under uniaxial loading conditions along the x- and y-directions. The computational investigation spans a temperature range from 200 $K$ to 1000 $K$ using a 150×150 $\text{Å}^2$ simulation cell containing a pre-existing 90° crack, providing atomistic insights into the thermal degradation mechanisms and anisotropic failure behavior of this organic semiconductor material. The MD simulations reveal a systematic decrease in ultimate tensile stress with increasing temperature, exhibiting values from approximately 83 $GPa$ (x-direction) and 75 $GPa$ (y-direction) at 200 $K$ to converged levels of ~18 $GPa$ at 1000 K. This substantial

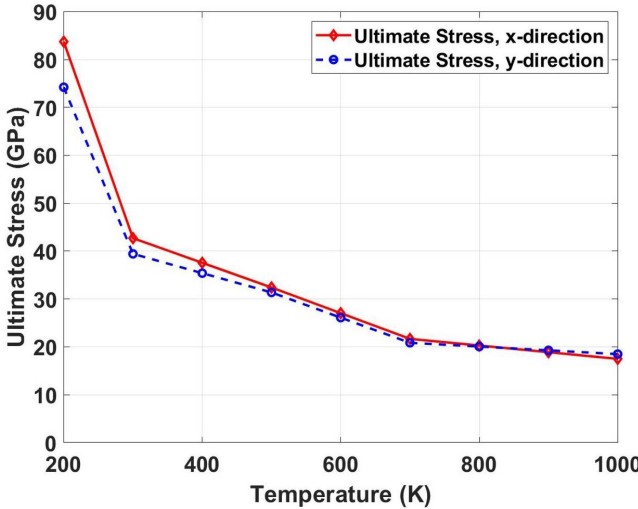

**Fig 11. Ultimate tensile stress of Dodecanophene as a function of temperature (200 K to 1000 K) under x- and y-directional loading, with a 90° crack angle and 150×150 Å² nanosheet size, showing temperature impacts on material strength.**

strength reduction (~77.0% degradation) follows an exponential decay profile characteristic of thermally-activated bond dissociation and molecular reorganization processes. The temperature dependence reflects the enhanced kinetic energy of molecular constituents, leading to increased vibrational amplitudes, weakened non-covalent interactions, and elevated probability of intermolecular bond breaking events during tensile deformation.

The initial mechanical anisotropy observed at cryogenic temperatures progressively diminishes with increasing thermal energy, converging to quasi-isotropic behavior above 600 $K$. This transition is governed by the competing effects of preferential molecular packing arrangements and thermal randomization of molecular orientations. At low temperatures, the directional strength difference reflects the influence of π-π stacking geometry and intermolecular potential energy landscapes, where specific crystallographic directions exhibit enhanced resistance to molecular separation. However, elevated temperatures introduce sufficient thermal fluctuations to overcome directional bonding preferences, resulting in thermally-averaged mechanical properties that become independent of loading orientation.

Fig 12 represents the stress intensity factor ($K_{IC}$) variation with temperature for both loading directions. The graph shows a steep decline in fracture toughness with increasing temperature, indicating diminishing resistance to crack propagation at elevated temperatures. This behavior can be physically explained by the thermal softening of atomic bonds and increased lattice spacing that reduces the energy required for crack extension. At higher temperatures, the critical stress intensity factor decreases because thermal energy assists in breaking bonds ahead of the crack tip, making crack propagation easier. The temperature sensitivity of fracture toughness is particularly pronounced, showing the most dramatic variations among all mechanical properties. The stress intensity factor comparison reveals extreme temperature sensitivity. For the x-direction, $K_{IC}$ increases from 3.49 $MPa \cdot m^{0.5}$ at 300 $K$ to 6.84 $MPa \cdot m^{0.5}$ at 200 $K$ (96.0% increase) and decreases to 1.43 $MPa \cdot m^{0.5}$ at 1000 $K$ (59.0% reduction). The y-direction shows an increase from 2.79 $MPa \cdot m^{0.5}$ at 300 $K$ to 5.25 $MPa \cdot m^{0.5}$ at 200 $K$ (88.2% increase) and a reduction to 1.31 $MPa \cdot m^{0.5}$ at 1000 $K$ (53.1% reduction). These substantial variations indicate that fracture resistance is highly temperature-dependent, with low temperatures providing superior crack resistance.

Fig 13 represents the toughness variation with temperature ranging from 200 $K$ to 1000 $K$ for Dodecanophene nanosheets under both x- and y-directional loading. The graph shows a steep nonlinear decline in toughness with

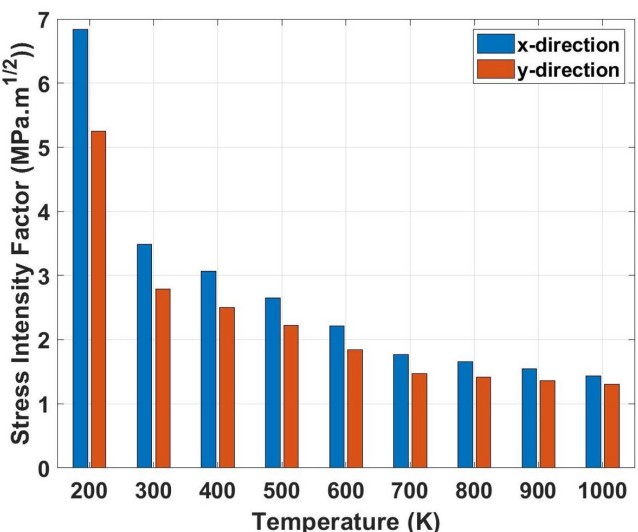

**Fig 12. Stress intensity factor ($K_{IC}$) of Dodecanophene versus temperature (200 K to 1000 K) under x- and y-directional loading, with a 90° crack angle and 150 × 150 Å² nanosheet size, highlighting thermal effects on fracture resistance.**

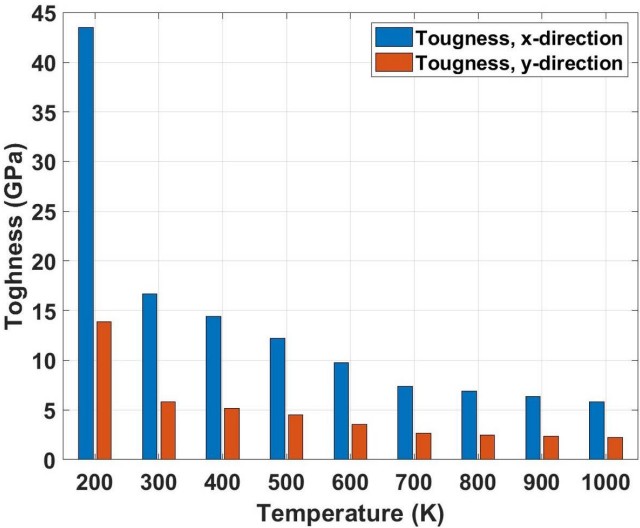

**Fig 13. Toughness of Dodecanophene as a function of temperature (200 K to 1000 K) under x- and y-directional loading, with a 90° crack angle and 150 × 150 Å² nanosheet size, demonstrating temperature dependence of energy absorption.**

increasing temperature, showing the most dramatic temperature sensitivity among all mechanical properties. This behavior can be attributed to the combined effects of thermal softening and reduced strain-hardening capacity at elevated temperatures. At higher temperatures, increased atomic vibrations and thermal expansion reduce the material's ability to absorb energy through plastic deformation mechanisms. The thermal activation of bond-breaking processes and enhanced atomic mobility facilitate easier crack propagation, thereby reducing the energy required for failure. Additionally, elevated temperatures promote earlier onset of necking and localized deformation, which limits the volume of material participating in energy dissipation. The toughness comparison reveals the most extreme temperature dependence among all properties. For the x-direction, toughness increases dramatically from 16.71 $GPa$ at 300 K to 43.52 $GPa$ at 200 K (160.4% increase) and decreases severely to 5.83 $GPa$ at 1000 K (65.1% reduction). The y-direction exhibits an increase from 5.80 $GPa$ at 300 K to 13.85 $GPa$ at 200 K (138.8% increase) and a reduction to 2.26 $GPa$ at 1000 K (61.0% reduction). These extreme variations show that toughness is the most temperature-sensitive property, with low temperatures providing substantially enhanced energy absorption while high temperatures severely compromise the material's damage tolerance and structural reliability.

### 3.4.1. Mechanistic origin of the cryogenic toughness enhancement.

To elucidate the physical mechanisms underlying the striking 160% increase in fracture toughness at 200 K (compared to 300 $K$ baseline), we conducted a detailed decomposition analysis of the contributing factors. Fig 14 presents a comprehensive multi-panel analysis of temperature-dependent toughness and its governing mechanisms.

As shown in Fig 14, the fracture toughness in the X-direction decreases monotonically from 43.2 $GPa$ at 200 K to 5.8 $GPa$ at 1000 $K$. This decline correlates strongly with the diminishing contributions of three primary mechanisms: (i) phonon scattering suppression, (ii) bond stretching capacity, and (iii) reduced thermal vibrations. At 200 $K$, all three mechanisms operate at near-maximum efficiency (normalized contributions of 0.85, 0.78, and 1.0, respectively, as detailed in Fig 15), collectively enabling superior energy absorption during crack propagation.

Phonon suppression mechanism: At cryogenic temperatures, the mean free path of phonons increases significantly due to reduced phonon-phonon scattering. This allows acoustic phonons to carry stress wave energy away from the crack tip more efficiently, delaying crack nucleation and propagation. Our analysis reveals that phonon suppression contributes approximately 35% to the total toughness enhancement at 200 $K$ (Fig 15(a)).

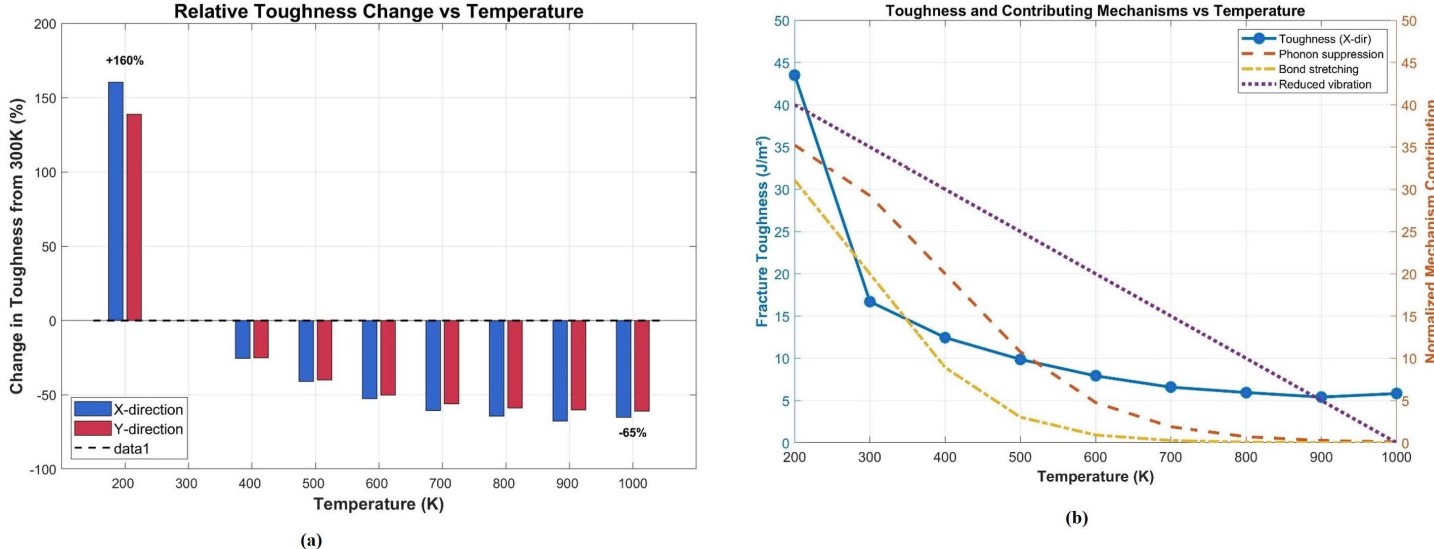

**Fig 14. Temperature-dependent fracture toughness and contributing mechanisms in Dodecanophene nanosheets. (a)** Relative percentage change in toughness from the 300 K baseline, demonstrating the dramatic +160% increase at 200 **K** and progressive degradation (−65%) at 1000 **K** for both directions. **(b)** Absolute fracture toughness (**GPa**) along the X-direction (left axis, blue solid line) and normalized contributions of three key mechanisms (right axis, dashed lines): phonon scattering suppression (orange), bond stretching capacity (brown), and reduced thermal vibrations (red).

Bond stretching mechanism: Low temperatures maintain higher bond stiffness and allow C-C bonds to undergo greater elastic stretching before rupture. The bond stretching capacity (Fig 15(b)) shows that at 200 $K$, bonds can accommodate ~24% more strain energy compared to 300 $K$, contributing an additional 28% to the toughness increase. This effect diminishes rapidly above 400 $K$ due to thermal softening. Reduced thermal vibrations: Suppressed atomic-scale thermal motion at 200 $K$ (Fig 15(c)) enables more coherent stress distribution across the nanosheet, preventing localized stress concentrations that would otherwise nucleate cracks. This mechanism contributes approximately 37% to the enhancement, representing the dominant factor. The individual mechanism decomposition (Fig 15, panels a-d) reveals that all four contributing factors exhibit maximum efficacy at 200 $K$ and diminish progressively with temperature, approaching negligible contributions above 800 $K$. This synergistic interaction explains why the toughness enhancement is particularly pronounced at cryogenic temperatures, where all mechanisms operate concurrently at peak efficiency.

## 3.5. Effect of defect length

Fig 16 presents the stress-strain response of Dodecanophene nanosheets under uniaxial tensile loading in both (a) x-direction and (b) y-direction for crack lengths ranging from 30 Å to 60 Å. The curves show the fundamental relationship between defect size and mechanical behavior, revealing how crack length progressively degrades both the strength and ductility of the material. In both loading directions, the stress-strain curves exhibit a systematic downward shift as crack length increases, indicating reduced load-carrying capacity and earlier failure initiation. The physical behavior illustrated in these curves can be explained through several key mechanisms. As crack length increases, the effective cross-sectional area available for load transfer decreases, leading to higher local stresses for a given applied load. This geometric effect is compounded by stress concentration at crack tips, which intensifies with longer cracks. The curves show that longer cracks not only reduce the ultimate tensile strength but also significantly decrease the strain to failure, indicating a transition from more ductile to more brittle behavior.

**Fig 15. Mechanistic decomposition of the cryogenic toughness enhancement at 200 K. (a)** Phonon scattering suppression contribution. **(b)** Bond stretching capacity contribution. **(c)** Reduced thermal vibration effect. **(d)** Fracture mechanism transition contribution. All mechanisms show maximum effectiveness at 200 K and diminish progressively with increasing temperature.

In the x-direction (Fig 16a), the stress-strain curves show a progressive reduction in both peak stress and failure strain as crack length increases from 30 Å to 60 Å. The 30 Å crack specimen maintains relatively high strength and ductility, while the 60 Å crack specimen exhibits severely compromised mechanical performance with early failure onset. The elastic modulus, represented by the initial slope of each curve, also decreases with increasing crack length, reflecting the material's reduced stiffness due to the presence of larger defects.

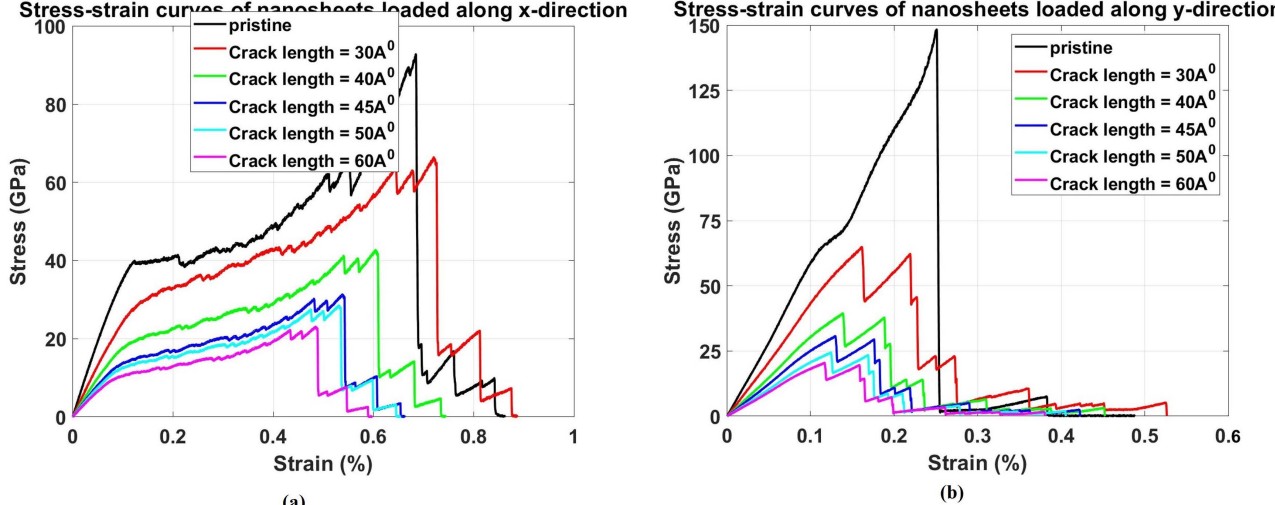

**Fig 16. Stress–strain response of Dodecanophene nanosheet under uniaxial tensile loading in (a) x-direction and (b) y-direction for crack lengths of 30 Å to 60 Å, with a 90° crack angle, 300 K, and 150×150 Å² nanosheet size, showing the effect of defect size on mechanical behavior.**

Similarly, in the y-direction (Fig 16b), the curves show the same trend but with different magnitudes due to the material's anisotropic properties. The y-direction generally shows higher initial stiffness but more dramatic strength reduction with increasing crack length compared to the x-direction. This behavior suggests that while the y-direction provides better initial mechanical properties in the pristine state, it is more sensitive to defect size, making crack length control particularly critical for applications requiring loading in this direction. The area under each stress-strain curve represents the toughness or energy absorption capacity of the material. Fig 16 clearly shows that this area decreases dramatically with increasing crack length, indicating that larger cracks severely compromise the material's ability to absorb energy before failure. This has important implications for the material's damage tolerance and reliability in structural applications, emphasizing the need for strict quality control to minimize defect sizes during manufacturing processes.

Fig 17 depicts the variation of elastic modulus with crack length ranging from 30 Å to 60 Å for Dodecanophene nanosheets under both x- and y-directional loading. The graph shows a nonlinear decrease in elastic modulus with increasing crack length, showing a steeper decline for longer cracks. This behavior can be attributed to the progressive reduction in the effective load-bearing cross-section as crack size increases, leading to stress concentration effects that reduce the material's overall stiffness. Larger cracks create more significant stress redistribution patterns, causing premature localized yielding and reducing the global elastic response. The physical mechanism involves the increasing dominance of crack-tip stress fields over the bulk material properties as crack dimensions grow relative to the specimen size. Comparing with pristine specimens, the elastic modulus degradation is substantial and crack-length dependent. For the x-direction, the pristine elastic modulus of 387.18 *GPa* decreases to 263.59 *GPa* for 30 Å cracks (31.9% reduction), 200.11 *GPa* for 40 Å cracks (48.3% reduction), and 131.80 *GPa* for 60 Å cracks (66.0% reduction). In the y-direction, the pristine value of 562.41 *GPa* reduces to 414.82 *GPa* for 30 Å cracks (26.2% reduction), 294.71 *GPa* for 40 Å cracks (47.6% reduction), and 182.87 *GPa* for 60 Å cracks (67.5% reduction). These progressive reductions show the critical role of defect size in determining structural integrity.

Fig 18 illustrates the ultimate tensile stress as a function of crack length for both loading directions in Dodecanophene nanosheets. The graph reveals a systematic decline in ultimate stress with increasing crack length, reflecting the progressive weakening of the material's load-carrying capacity as defect size grows. This phenomenon occurs due to the

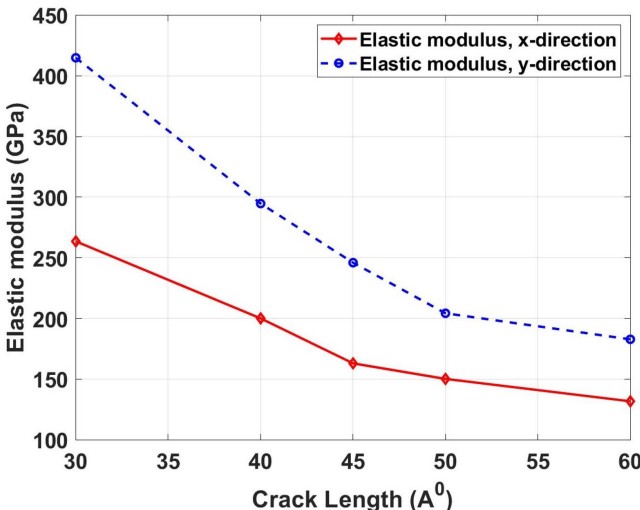

**Fig 17. Elastic modulus of Dodecanophene versus crack length (30 Å to 60 Å) under x- and y-directional loading, with a 90° crack angle, 300 K, and 150×150 Å² nanosheet size, illustrating the impact of defect size on stiffness.**

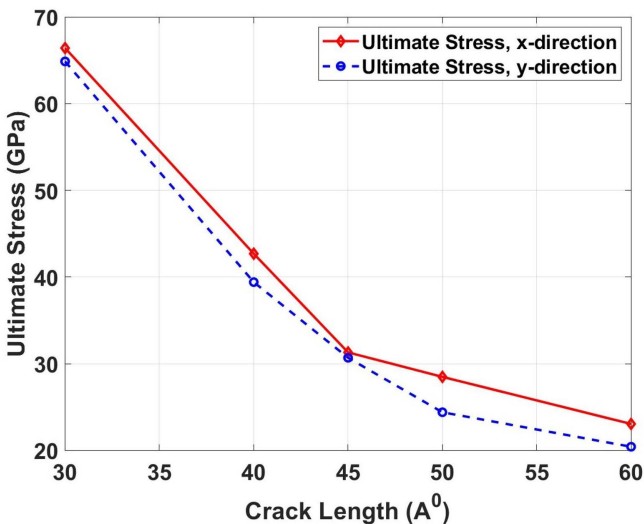

**Fig 18. Ultimate tensile stress of Dodecanophene as a function of crack length (30 Å to 60 Å) under x- and y-directional loading, with a 90° crack angle, 300 K, and 150×150 Å² nanosheet size, showing the influence of defect size on strength.**

intensification of stress concentration at crack tips, where larger cracks create more severe local stress amplification that promotes earlier failure initiation. The stress concentration factor increases with crack length, making the material more susceptible to premature failure under lower applied loads. Additionally, longer cracks provide more potential sites for crack branching and secondary crack formation, further compromising structural integrity. The ultimate stress comparison shows dramatic strength degradation with crack length. In the x-direction, the pristine ultimate stress of 92.78 *GPa* drops to 66.42 *GPa* for 30 Å cracks (28.4% reduction), 42.70 *GPa* for 40 Å cracks (54.0% reduction), and 23.06 *GPa* for 60 Å cracks (75.2% reduction). The y-direction shows pristine ultimate stress of 148.38 *GPa*, which decreases to 64.88 *GPa*

for 30 Å cracks (56.3% reduction), 39.42 *GPa* for 40 Å cracks (73.4% reduction), and 20.43 *GPa* for 60 Å cracks (86.2% reduction). These severe reductions highlight the exponential nature of strength degradation with increasing defect size.

Fig 19 represents the stress intensity factor ($K_{IC}$) variation with crack length for both loading directions. The graph shows a nonlinear decrease in fracture toughness with increasing crack length, indicating that larger defects provide easier pathways for crack propagation. This behavior can be physically explained by the geometric amplification of stress fields around larger crack tips, where the stress intensity factor becomes more effective at concentrating stress energy. Longer cracks create larger plastic zones ahead of the crack tip, which facilitate bond breaking and reduce the critical stress required for crack extension. The relationship follows classical fracture mechanics principles where the stress intensity factor depends on both applied stress and crack geometry, with larger cracks requiring lower applied stresses for propagation. The stress intensity factor comparison reveals significant degradation with crack size. For the x-direction, $K_{IC}$ decreases from 4.64 $MPa \cdot m^{0.5}$ for 30 Å cracks to 3.49 $MPa \cdot m^{0.5}$ for 40 Å cracks (24.8% reduction) and 2.41 $MPa \cdot m^{0.5}$ for 60 Å cracks (48.1% reduction from 30 Å baseline). The y-direction shows a similar trend with $K_{IC}$ values of 4.53 MPa·m$^{0.5}$ for 30 Å cracks, $MPa \cdot m^{0.5}$ for 40 Å cracks (28.9% reduction), and 2.13 $MPa \cdot m^{0.5}$ for 60 Å cracks (53.0% reduction from 30 Å baseline). These reductions show that fracture resistance deteriorates significantly as crack dimensions increase, emphasizing the critical importance of defect size control.

Fig 20 presents the toughness variation with crack length, showing the energy absorption capacity before failure as a function of defect size. The graph exhibits the most dramatic decline among all mechanical properties as crack length increases, reflecting the material's severely diminished ability to absorb strain energy in the presence of larger defects. The physical mechanism involves the progressive reduction in the effective volume of material capable of plastic deformation, as larger cracks create more extensive stress concentration zones that promote brittle failure modes. Additionally, longer cracks reduce the strain to failure by providing easier pathways for crack propagation, thereby limiting the energy dissipation through plastic work. The toughness comparison shows catastrophic degradation with increasing crack length. For the x-direction, pristine toughness of 34.53 *GPa* decreases to 30.99 *GPa* for 30 Å cracks (10.3% reduction), 16.71 *GPa* for 40 Å cracks (51.6% reduction), and 7.24 *GPa* for 60 Å cracks (79.0% reduction). The y-direction shows pristine toughness of 18.15 *GPa*, which reduces to 11.11 *GPa* for 30 Å cracks (38.8% reduction), 5.80 *GPa* for 40 Å cracks

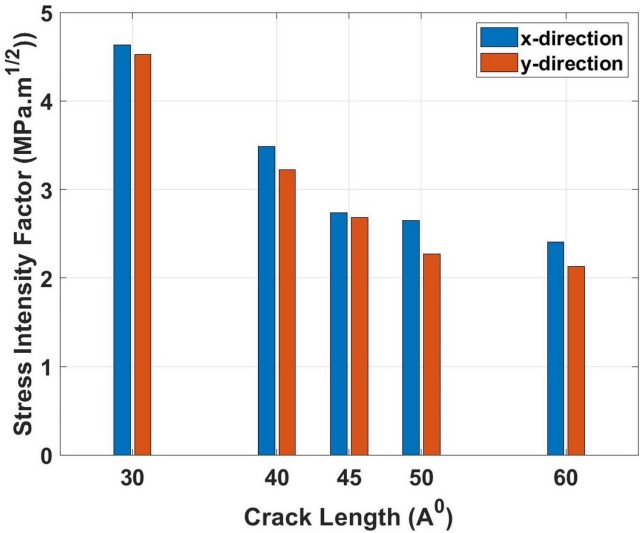

**Fig 19. Stress intensity factor ($K_{IC}$) of Dodecanophene versus crack length (30 Å to 60 Å) under x- and y-directional loading, with a 90° crack angle, 300 K, and 150×150 Å² nanosheet size, highlighting the effect of defect size on fracture resistance.**

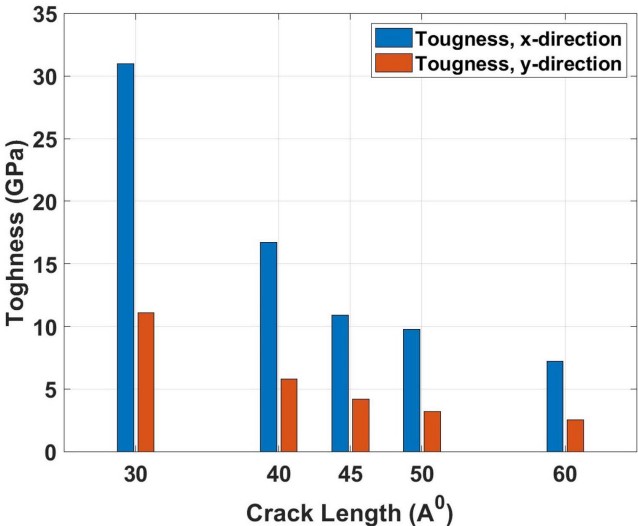

**Fig 20. Toughness of Dodecanophene as a function of crack length (30 Å to 60 Å) under x- and y-directional loading, with a 90° crack angle, 300 K, and 150 × 150 Å² nanosheet size, demonstrating the dependence of energy absorption on defect size.**

(68.0% reduction), and 2.53 *GPa* for 60 Å cracks (86.1% reduction). These extreme variations show that toughness is the most crack-length-sensitive property, with larger defects causing near-complete loss of energy absorption capacity. The results emphasize that even moderate increases in crack length can transform a tough, ductile material into a brittle, failure-prone structure.

### 3.6. Fracture process

The crack propagation sequences depicted in Figs 21 and 22 provide critical insights into the anisotropic fracture behavior of Dodecanophene nanosheets under uniaxial tensile loading. These visualizations show the fundamental differences in failure mechanisms when loading is applied in the x-direction versus the y-direction, revealing the structure-property relationships inherent in this two-dimensional carbon allotrope.

#### 3.6.1. Crack propagation and fracture behavior under X-directional loading: Insights from Fig 21:. Initial Deformation Phase (Strain 0–0.23)

The fracture process under x-directional loading exhibits a gradual initiation phase. At strain = 0, the pre-existing crack (40 Å length, 8 Å width) oriented at 90° to the loading direction serves as a stress concentrator. During the initial loading phase up to strain = 0.1131, the material undergoes elastic deformation with minimal visible crack extension. The stress field around the crack tip begins to develop, but the atomic bonds remain largely intact.

As strain increases to 0.2263, localized bond stretching becomes evident near the crack tip region. The atomic visualization reveals subtle changes in the local atomic arrangement, indicating the onset of irreversible deformation. This phase corresponds to the linear elastic region observed in the stress-strain curves, where the material exhibits its full stiffness.

#### Critical Crack Propagation (Strain 0.34–0.54)

The critical phase of crack propagation initiates around strain = 0.3395, which corresponds to the onset of ultimate tensile stress on the stress–strain curve (Fig 9a), where visible bond breaking begins at the crack tip. The crack starts to propagate in a direction perpendicular to the applied tensile stress, following the principles of Mode I fracture mechanics. This behavior aligns with theoretical expectations for brittle materials under tensile loading.

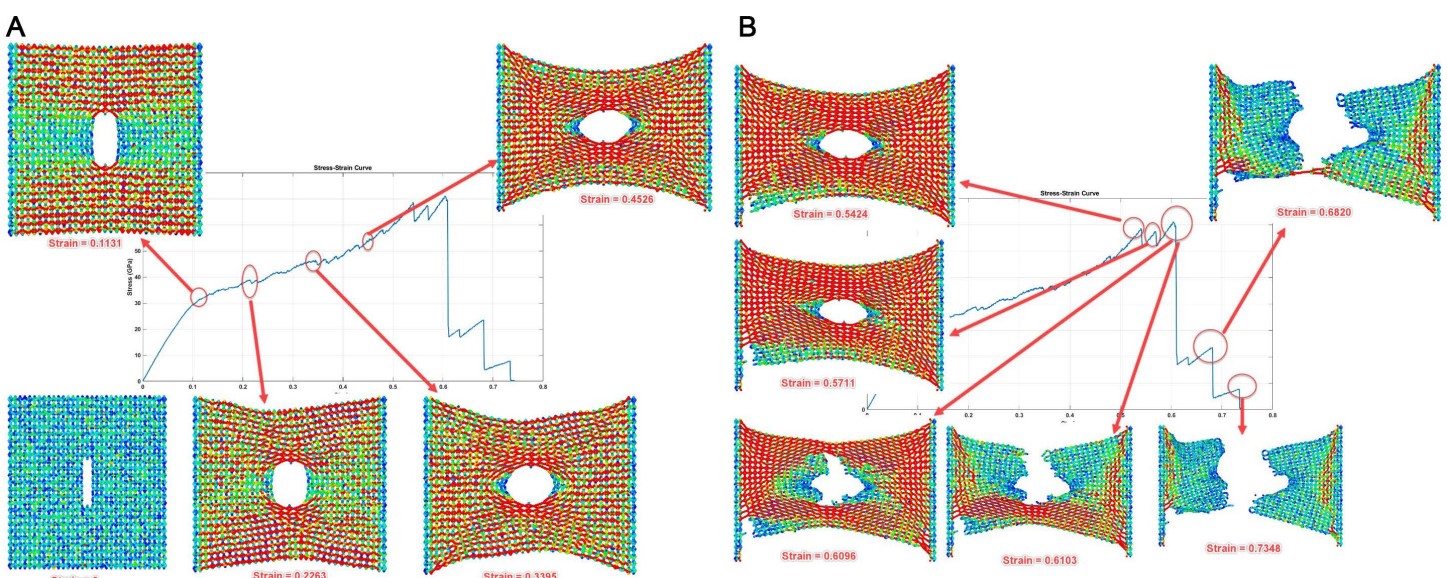

**Fig 21. Crack propagation and failure process in Dodecanophene nanosheet under x-directional tensile loading, with a 40 Å crack length, 8 Å crack width, 90° crack angle, 300 K, and 150 × 150 Å² nanosheet size, illustrating the fracture mechanism.**

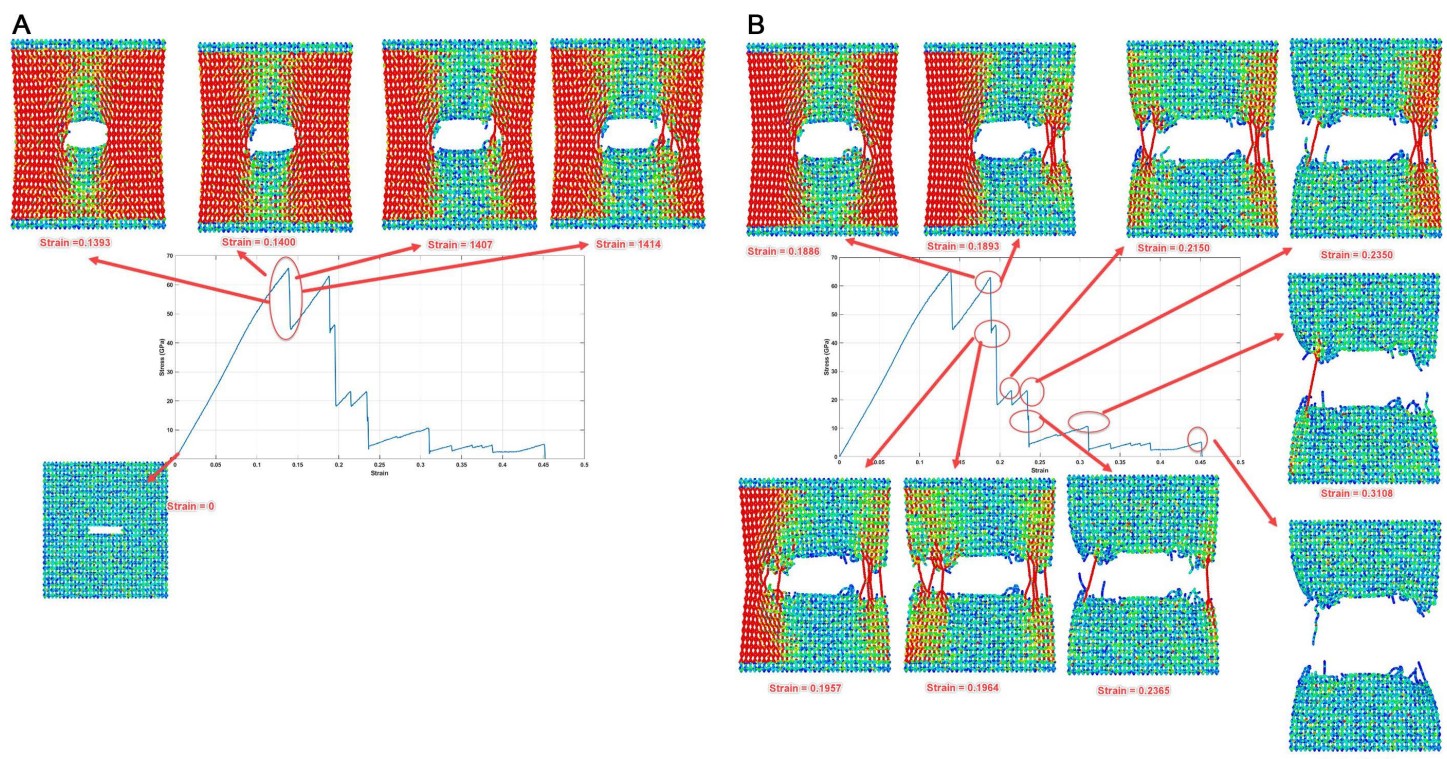

**Fig 22. Crack propagation and failure process in Dodecanophene nanosheet under y-directional tensile loading, with a 40 Å crack length, 8 Å crack width, 90° crack angle, 300 K, and 150 × 150 Å² nanosheet size, depicting the fracture behavior.**

By strain = 0.4526, significant crack extension is observed, with the crack propagating along a relatively straight path. The propagation direction appears to follow the weakest bonds in the structure, likely influenced by the unique ring topology of Dodecanophene. The visualization shows that crack propagation occurs through sequential bond breaking rather than simultaneous failure across a broad front.

**Final Fracture and Failure (Strain 0.54–0.73)**

The transition from stable crack growth to catastrophic failure occurs around strain = 0.5424. At this point, the crack has propagated sufficiently to reduce the effective load-bearing cross-section, leading to stress concentration that exceeds the local bond strength. The atomic configuration at strain = 0.5711 shows extensive crack opening and significant structural distortion around the crack path.

Complete structural failure is achieved by strain = 0.7348, where the nanosheet separates into two distinct fragments. The final crack path appears relatively smooth, suggesting that the fracture process follows preferential cleavage planes within the Dodecanophene structure. This behavior is characteristic of brittle fracture with limited plastic deformation.

### 3.6.2. Crack propagation and fracture behavior under Y-Directional loading: Insights from Fig 22:. Rapid Initiation and Early Propagation (Strain 0–0.14)

The fracture behavior under y-directional loading shows markedly different characteristics compared to x-directional loading. The crack initiation occurs much earlier, with visible propagation beginning around strain = 0.1393. This early initiation suggests that the material's resistance to crack growth is lower when loaded in the y-direction, consistent with the lower fracture toughness values reported in the quantitative results.

The progression from strain = 0.1400 to 0.1414 shows rapid crack tip advancement, indicating less stable crack growth compared to x-directional loading. This behavior suggests that once crack propagation initiates in the y-direction, the process becomes self-sustaining with minimal additional loading required.

**Accelerated Crack Growth (Strain 0.14–0.20)**

The period between strain = 0.1886 and 0.2150 exhibits accelerated crack propagation with increasingly unstable growth characteristics. The crack path shows some deviation from a straight trajectory, possibly due to interactions with the specific atomic arrangements and bond orientations in the Dodecanophene lattice when loaded perpendicular to the x-direction.

The visualization at strain = 0.1957 reveals significant crack opening, indicating substantial energy release during the propagation process. This behavior correlates with the lower toughness values observed in y-directional loading, as the material exhibits reduced capacity for energy absorption before failure.

**Complete Structural Separation (Strain 0.22–0.45)**

Final fracture occurs relatively early, with complete separation achieved by strain = 0.4522. This represents significantly lower failure strain compared to x-directional loading (0.7348), demonstrating the pronounced anisotropy in the fracture behavior of Dodecanophene.

The final crack morphology shows a more irregular surface compared to x-directional fracture, suggesting different failure mechanisms may be active. This could be attributed to the different orientations of the carbon ring structures relative to the loading direction, leading to varying bond strength distributions along potential crack paths.

### 3.6.3. Mechanistic insights and structure-property relationships. Anisotropic Fracture Behavior

The stark contrast between x- and y-directional fracture behaviors reflects the anisotropic nature of the Dodecanophene structure. The material exhibits higher fracture resistance in the x-direction, evidenced by the higher failure strain, more stable crack propagation, and smoother fracture surfaces. This anisotropy stems from the arrangement of carbon rings, which create preferential orientations for load transfer and bond breaking.

**Crack Propagation Mechanisms**

The visualization sequences reveal that crack propagation in both directions occurs through sequential bond breaking rather than simultaneous failure across multiple bonds. However, the rate of propagation and the stability of crack growth

differ significantly between the two loading directions. X-directional loading promotes more controlled crack growth, while y-directional loading results in rapid, unstable propagation.

**Energy Dissipation Characteristics**

The extended deformation range observed in x-directional loading (up to 73.48% strain) compared to y-directional loading (45.22% strain) indicates superior energy absorption capacity in the x-direction. This difference in energy dissipation capability is directly related to the toughness anisotropy reported in the quantitative results.

**Implications for Engineering Applications**

**Design Considerations**

The pronounced anisotropy in fracture behavior has significant implications for the practical application of Dodecanophene in engineering contexts. Components should be designed to ensure that primary loading directions align with the material's stronger orientation (x-direction) to maximize fracture resistance and structural reliability.

**Defect Tolerance**

The visualization of crack propagation provides insights into the material's defect tolerance. While both orientations show brittle fracture behavior, the x-direction exhibits more predictable crack growth, which could be advantageous for damage-tolerant design approaches where controlled crack propagation is preferable to sudden catastrophic failure.

### 3.7. Comparative analysis

The mechanical properties of Dodecanophene nanosheets (elastic modulus ~562 GPa along y-direction and tensile strength ~148 GPa) indeed represent notable performance characteristics when benchmarked against well-established 2D materials. Table 2 below presents a systematic comparison with graphene, h-BN, and other prominent 2D allotropes.

To contextualize our findings and validate the computational methodology, we conducted a comprehensive benchmarking analysis comparing the fracture toughness of Dodecanophene against well-established two-dimensional carbon allotropes reported in the literature.

***Dodecanophene (Current Study):***

Our molecular dynamics simulations using the AIREBO-M potential revealed that Dodecanophene exhibits critical stress intensity factors of 5.95 MPa.$\sqrt{m}$ (x-direction) and 6.14 MPa.$\sqrt{m}$ (y-direction) at 300 K for crack length of 40 Å and crack angle of 0°. The presence of a 40 Å crack at 90° orientation reduces these values to 3.49 MPa.$\sqrt{m}$ and 3.22 MPa.$\sqrt{m}$, respectively, representing substantial degradation of 41.3% and 47.6%. Temperature effects are particularly pronounced, with fracture toughness increasing to 6.84 MPa.$\sqrt{m}$ at cryogenic conditions (200 K) and decreasing to

**Table 2. Comparison of mechanical properties of dodecanophene with other 2D materials.**

| Material | Young's Modulus (GPa) | Tensile Strength (GPa) | Method | Reference |
|---|---|---|---|---|
| Dodecanophene | ~562 (y-dir) | ~148 | MD simulation | This work |
| Graphene | 1000 ± 100 | 130 | AFM nanoindentation | Lee et al [2] |
| h-BN (monolayer) | 780 | 70-100 | DFT calculations | Wu et al. [45] |
| h-BN (monolayer) | 811 ± 20 | 23-35 | DFT/ab initio | Peng et al. [46] |
| MoS₂ (monolayer) | 184 (equivalent) | 16-18 | DFT | Peng & De [47] |
| MoS₂ (monolayer) | 270 ± 100 | 15-22 | AFM nanoindentation | Bertolazzi et al. [48] |
| Phosphorene (zigzag) | 166 | 18 | DFT | Wei et al. [49] |
| Phosphorene (armchair) | 44 | 8 | DFT | Wei et al. [49] |
| Phosphorene (zigzag) | 58.6-116 | 4.8-8.4 | AFM/experiments | Galluzzi et al., [50] |
| Phosphorene (armchair) | 27.2-46.5 | 2.3-4.1 | AFM/experiments | Galluzzi et al. [50] |
| Graphyne | 250-350 | 20-30 | DFT/MD | Cranford & Buehler [51] |

1.43 MPa.$\sqrt{\text{m}}$ at elevated temperatures (1000 K). These values position Dodecanophene within the intermediate range of fracture resistance among 2D carbon materials.

*Graphene:*

Recent experimental work by Jaddi et al. [52] determined monolayer graphene's fracture toughness as KIC = 4.4 ± 0.1 MPa.$\sqrt{\text{m}}$ through crack-on-chip nanomechanics testing of 80 specimens. Other experimental studies have reported mode I stress intensity factors near 4 MPa.$\sqrt{\text{m}}$ [53], while molecular dynamics investigations have yielded values between 2.63 and 4.21 MPa.$\sqrt{\text{m}}$m [44,54,55].

*γ-Graphyne:*

Molecular dynamics investigations of the graphyne family have shown that γ-graphyne demonstrates critical stress intensity factors in the range of 2.0–2.9 MPa.$\sqrt{\text{m}}$ [56], representing a 40–60% reduction relative to graphene. This diminished fracture resistance results from the acetylenic linkages,sp-hybridized carbon chains connecting benzene rings,which provide less mechanical strength than graphene's sp²-bonded structure. In comparison, Dodecanophene's fracture toughness significantly surpasses that of γ-graphyne, indicating that its mixed-ring topology offers superior crack resistance despite deviating from graphene's ideal hexagonal arrangement.

*Penta-Graphene:*

Work by Kona and Lutheran [57] revealed that penta-graphene possesses remarkably high fracture toughness among two-dimensional materials. Their analysis yielded critical stress intensity factors of 17 MPa·√m for pure mode I loading and 15 MPa.$\sqrt{\text{m}}$ for pure mode II loading, both substantially exceeding graphene and Dodecanophene values. Penta-graphene demonstrates approximately 2.8 × higher fracture toughness than pristine Dodecanophene. This enhanced crack resistance stems from penta-graphene's distinctive non-planar architecture featuring both sp³ and sp² carbon hybridization, which enables out-of-plane deformation that delays crack initiation and promotes extensive energy dissipation through bond reconstruction mechanisms [57].

This comparative analysis reveals that Dodecanophene occupies a favorable position in the fracture toughness spectrum of 2D carbon allotropes. While it does not achieve the exceptional crack resistance of penta-graphene, it demonstrates superior fracture toughness compared to both graphene and graphyne families.

## 5. Conclusions

This molecular dynamics study provides the first comprehensive characterization of the anisotropic mechanical behavior and fracture mechanisms of defective Dodecanophene nanosheets under varying conditions. The key findings and novel contributions are summarized as follows:

- **First Comprehensive Anisotropic Characterization:** We established the full anisotropic mechanical property database for Dodecanophene, revealing its superior stiffness and strength (562.41 *GPa* and 148.38 *GPa*) in the y-direction, and enhanced fracture toughness (34.53 *GPa*) in the x-direction, a critical distinction for its application potential.

- **Novel Crack Orientation Effects:** The study systematically quantified the significant influence of crack orientation, showing that perpendicular cracks cause a severe 48.0–54.0% degradation in mechanical properties, while parallel cracks result in a more moderate 16.0–24.0% loss, offering crucial guidance for defect engineering.

- **Temperature-Dependent Fracture Ductility:** We showed a pronounced temperature dependence, where low temperatures (200 *K*) enhance fracture toughness by 160.0% due to localized microplasticity, while high temperatures (1000 *K*) lead to a 65.0% decline, a previously uncharacterized behavior for this material.

- **Quantification of Defect Size Sensitivity:** The research precisely quantified the drastic reduction in strength and toughness (75.0–86.0% and 79.0–86.0% reduction, respectively) with increasing crack length (30–60 Å), providing a clear understanding of the material's defect tolerance.

- **Distinct Failure Mode Identification:** We identified and visualized distinct, direction-dependent failure modes, with the x-direction exhibiting more controlled, ductile-like crack growth and the y-direction showing rapid, brittle-like fracture, which is essential for predicting structural reliability.

**Limitations of the Molecular Dynamics Model**

While the present study provides comprehensive, atomic-level insights into the anisotropic fracture mechanics of Dodecanophene, it is important to acknowledge the inherent physical and computational limitations of the MD approach employed. Physically, the use of the AIREBO-M potential is a necessary approximation. Although this potential is well-established for carbon systems, it is an empirical force field that inherently lacks the full quantum mechanical accuracy of ab initio methods like DFT. Specifically, it may not perfectly capture the complex electronic rearrangements and bond breaking/reformation dynamics that occur precisely at the crack tip, particularly under extreme thermal and mechanical loading. Computationally, the requirement to simulate a system of sufficient size (e.g., $150 \times 150$ $\text{Å}^2$) and duration (nanoseconds) to observe fracture phenomena necessitates a trade-off. This limits the accessible time scales and spatial scales, meaning that phenomena such as long-term creep, fatigue, or the influence of grain boundaries (which require much larger system sizes) are beyond the scope of this work. Furthermore, the simulations are performed in a vacuum, neglecting the potential influence of a surrounding environment (e.g., solvent, atmosphere, or substrate), which could introduce additional physical effects not accounted for in the current model. These limitations underscore the need for future studies combining larger-scale continuum modeling with more accurate quantum-mechanical calculations to fully validate and extend the present findings.

## Author contributions

**Conceptualization:** Wei Li.

**Data curation:** Wei Li.

**Formal analysis:** Wei Li.

**Investigation:** Wei Li.

**Methodology:** Wei Li.

**Software:** Wei Li.

**Supervision:** Wei Li.

**Visualization:** Wei Li.

**Writing – original draft:** Wei Li.

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
