## [Decision Letter · Decision Letter 0]

13 Oct 2025

Dear Dr. Li,

Thank you for submitting your manuscript to PLOS ONE. After careful consideration, we feel that it has merit but does not fully meet PLOS ONE’s publication criteria as it currently stands. Therefore, we invite you to submit a revised version of the manuscript that addresses the points raised during the review process.

We look forward to receiving your revised manuscript.

Kind regards,

Mohammad Azadi

Academic Editor

PLOS ONE

Additional Editor Comments:

The manuscript must be revised based on the reviewers’ comments plus the following issues,

1) A separated file must be provided for the authors’ answers to the comments, one by one. Moreover, all changes must be yellow-colored highlighted sentences in the revised article. The track changes condition is not suggested.

2) Using references suggested by the reviewers is not mandatory. If they are related, they could be used by the authors. If not, they could be ignored by the authors.

3) It is not clear what the novelty is in the article title.

4) All keywords must be found in the abstract or the title.

5) 10 references for the first paragraph of the introduction. It must be shortened.

6) Compare your manuscript and the following publications in your introduction.

*https://scholar.google.com/scholar?hl=en&as_sdt=0%2C5&q=Thermal-+and+Defect-Induced+Mechanical+Behavior+of+Dodecanophene++Nanosheets&btnG=

7) The novelty of the manuscript must be highlighted in the introduction, compared to the literature review.

8) Generally, the introduction is lengthy. Only 2-page is enough.

9) 43 references are used for the introduction. Almost all references are in this section. However, they must be used for a better discussion. Using 20-25 references for the introduction and 20-25 references for the discussion is good.

10) All formulations need references, unless they were extracted or introduced by the authors.

11) All used material properties need references.

12) The validation section must be extended and it is better to report in a table to easily find the error.

13) The discussion is poor and it must be improved. They must be compared to other results of other similar articles.

14) The conclusions section should be rewritten one by one, in bullets, to show the novelty. It is also lengthy and it must be shortened.

15) References should be updated based on recent articles, published in 2015-2025. Moreover, it should be extended to at least 35 articles for a proper discussion.

Reviewers' comments:

Reviewer's Responses to Questions

**Comments to the Author**

1. Is the manuscript technically sound, and do the data support the conclusions?

Reviewer #1: Yes

Reviewer #2: Partly

2. Has the statistical analysis been performed appropriately and rigorously?

Reviewer #1: N/A

Reviewer #2: Yes

3. Have the authors made all data underlying the findings in their manuscript fully available?

Reviewer #1: Yes

Reviewer #2: No

4. Is the manuscript presented in an intelligible fashion and written in standard English?

Reviewer #1: Yes

Reviewer #2: No

Reviewer #1: The manuscript presents a well-structured and comprehensive molecular dynamics (MD) investigation of Dodecanophene nanosheets, focusing on anisotropic mechanical behavior under varying defect sizes, crack orientations, and temperatures. The study is thorough, data-rich, and effectively contextualized within the literature on 2D carbon allotropes. The methodology and analysis are clear, with figures and results supporting the conclusions convincingly.

The paper is suitable for publication after major revisions to improve clarity, consistency, and minor stylistic details.

The manuscript does not specify the exact strain rate applied during the uniaxial tensile simulations. Since the deformation rate strongly affects stress–strain behavior in molecular dynamics (due to inertial effects and limited timescales), the omission leaves ambiguity regarding the quantitative reliability and transferability of the results to quasi-static loading conditions.

The critical stress intensity factor, image.png, which originates from linear elastic fracture mechanics (LEFM) developed for macroscale brittle materials. However, Dodecanophene is a nanoscale system where atomic discreteness and lattice trapping effects dominate. It is only meaningful for Mode I (90° cracks) and does not rigorously apply to mixed-mode (30°, 45°, 60°) fracture cases.

LEFM assumes a continuum, isotropic, and linear elastic material up to fracture initiation, which does not strictly hold for atomic lattices that exhibit bond breaking and reformation.

The manuscript lacks an explicit discussion section outlining the model’s physical and computational limitations. A short dedicated paragraph at the end of Section 4 (Conclusions) would make the study more balanced and credible.

Add a concise statement about the computational model validation (comparison with DFT) for completeness.

In Figure 9, the stress–strain curves at different temperatures reveal that, at 200 K, the Dodecanophene nanosheet with a pre-existing 90° crack exhibits a slightly higher fracture strain than the pristine structure. Need more detail description.

Replace phrases such as “These comprehensive molecular dynamics study” → “This comprehensive molecular dynamics study.”

Use “shows” instead of “demonstrates” in repetitive cases to improve readability.

Replace “the figure shows” → “Figure X illustrates” for conciseness.

Units for mechanical quantities should be uniform:

Replace “MPa” with “GPa” in Fig. 3 description to match the reported magnitudes.

Ensure the reported percentage reductions are rounded consistently (e.g., one decimal place).

Verify figure numbering: “Figures 19 and 30” (page 25) should likely be “Figures 19 and 20.”

Some figures (e.g., Figs. 9–13) could benefit from slightly larger axis fonts and consistent scaling for easy comparison.

Clarify whether all crack simulations were performed at 300 K except when temperature effects were explicitly tested.

In Section 3.6.1, “visible bond breaking begins at the crack tip at strain = 0.3395” — specify if this corresponds to the onset of ultimate stress on the stress–strain curve.

Replace text-based equations with equation editor formatting where applicable.

Reviewer #2: 1- The abstract states "distinct failure modes emerging," but doesn't describe them. The abstract is too long and lacks an introduction or innovation.

2- The results are too long and should be summarized effectively.

3- The language needs improvement.

4- How do these properties (e.g., ~562 GPa stiffness, ~148 GPa strength) compare to graphene, h-BN, or other 2D allotropes? Contextualizing the performance helps assess the potential of Dodecanophene.

5- The 160% toughness increase at 200 K is striking. Is this due to suppressed phonon scattering, a change in the dominant fracture mechanism, or reduced thermal vibrations allowing for more bond stretching?

6-While an abstract can't have details, a expert reader might wonder about the specific interatomic potential (force field) used and its validation for fracture simulations.

7-The introduction is too long and innovation should be applied in a summary.

8-The phrase "deviates from perpendicular alignment with the loading direction" is slightly ambiguous. It's technically correct, but a reader must pause to visualize it.

9-You state that the y-direction has "superior crack resistance" but don't explain why. Given that the y-direction was also the stiffer, stronger direction in the abstract, this is a critical point.

10-You mention the "reference calculation from pristine ultimate stress" without explanation. A expert reader might wonder about the method.

11-The abstract reported fracture toughness in J/m² (a measure of energy), while here it's in MPa·m⁰·⁵ (stress intensity). These are equivalent measures for linear elastic fracture mechanics, but making the connection explicit is helpful.

**Do you want your identity to be public for this peer review?** For information about this choice, including consent withdrawal, please see our Privacy Policy

Reviewer #1: No

Reviewer #2: No

---

## [Decision Letter · Decision Letter 1]

2 Dec 2025

Dear Dr. Li,

Thank you for submitting your manuscript to PLOS ONE. After careful consideration, we feel that it has merit but does not fully meet PLOS ONE’s publication criteria as it currently stands. Therefore, we invite you to submit a revised version of the manuscript that addresses the points raised during the review process.

We look forward to receiving your revised manuscript.

Kind regards,

Mohammad Azadi

Academic Editor

PLOS ONE

**Journal Requirements:**

**Additional Editor Comments:**

Still, the following issues must be addressed,

1) The keyword of "Molecular dynamics simulation" cannot be find in the title or the abstract. How can it be a keyword?

2) The whole abstract includes the results. Just the first sentence is the topic. The research method must be mentioned.

3) What is "DFT" in the abstract? All abbreviations need to be defined at first mentioning.

4) What is the reference for a constant engineering strain rate of 100,000 /s?

5) There is a section for 2.1 without any 2.2! How can it be?

6) Figure 3 is mentioned without any related text before it! Check all figures.

7) The title of Figure 15 is so lengthy. The description of figures must be mentioned in the main text. Check all figure titles.

8) The discussion is not just to describe the results. They must be compared to the other articles. Therefore, the discussion is still poor. This issue can be also understood from one reviewer's comment that Section 3.4.1 is added without any references.

9) I cannot see any highlighted references in the revised manuscript. No changes in references?

Reviewers' comments:

Reviewer's Responses to Questions

**Comments to the Author**

Reviewer #1: All comments have been addressed

Reviewer #3: All comments have been addressed

2. Is the manuscript technically sound, and do the data support the conclusions?

Reviewer #1: Yes

Reviewer #3: Yes

3. Has the statistical analysis been performed appropriately and rigorously?

Reviewer #1: Yes

Reviewer #3: Yes

4. Have the authors made all data underlying the findings in their manuscript fully available?

Reviewer #1: Yes

Reviewer #3: Yes

5. Is the manuscript presented in an intelligible fashion and written in standard English?

Reviewer #1: Yes

Reviewer #3: Yes

Reviewer #1: The authors have fully addressed to my comments. Therefore, my suggestion is acceptance of the paper in current form

Reviewer #3: (No Response)

**Do you want your identity to be public for this peer review?** For information about this choice, including consent withdrawal, please see our Privacy Policy

Reviewer #1: No

Reviewer #3: No

---

## [Editor Report · Decision Letter 2]

8 Dec 2025

Anisotropic Mechanical Properties of Dodecanophene Nanosheets with Pre-Existing Cracks by Molecular Dynamics Simulation: Uncovering Orientation- and Temperature-Induced Variations

PONE-D-25-51742R2

Dear Dr. Li,

We’re pleased to inform you that your manuscript has been judged scientifically suitable for publication and will be formally accepted for publication once it meets all outstanding technical requirements.

Kind regards,

Mohammad Azadi

Academic Editor

PLOS One

Additional Editor Comments (optional):

Almost done!
---

## [Editor Report · Acceptance letter]

PONE-D-25-51742R2

PLOS One

Dear Dr. Li,

I'm pleased to inform you that your manuscript has been deemed suitable for publication in PLOS One. Congratulations! Your manuscript is now being handed over to our production team.

Kind regards,

on behalf of

Dr. Mohammad Azadi

Academic Editor

PLOS One